# Ketamine's rapid antidepressant effects are mediated by Ca²⁺-permeable AMPA receptors

**Anastasiya Zaytseva[1†], Evelina Bouckova[1†], McKennon J Wiles[1†], Madison H Wustrau[2†], Isabella G Schmidt[1†], Hadassah Mendez-Vazquez[2], Latika Khatri[3], Seonil Kim[1,2]***

[1]Molecular, Cellular and Integrative Neurosciences Program, Colorado State University, Fort Collins, United States; [2]Department of Biomedical Sciences, Colorado State University,, Fort Collins, United States; [3]Department of Cell Biology, New York University Grossman School of Medicine, New York, United States

**Abstract** Ketamine is shown to enhance excitatory synaptic drive in multiple brain areas, which is presumed to underlie its rapid antidepressant effects. Moreover, ketamine's therapeutic actions are likely mediated by enhancing neuronal Ca²⁺ signaling. However, ketamine is a noncompetitive NMDA receptor (NMDAR) antagonist that reduces excitatory synaptic transmission and postsynaptic Ca²⁺ signaling. Thus, it is a puzzling question how ketamine enhances glutamatergic and Ca²⁺ activity in neurons to induce rapid antidepressant effects while blocking NMDARs in the hippocampus. Here, we find that ketamine treatment in cultured mouse hippocampal neurons significantly reduces Ca²⁺ and calcineurin activity to elevate AMPA receptor (AMPAR) subunit GluA1 phosphorylation. This phosphorylation ultimately leads to the expression of Ca²⁺-Permeable, GluA2-lacking, and GluA1-containing AMPARs (CP-AMPARs). The ketamine-induced expression of CP-AMPARs enhances glutamatergic activity and glutamate receptor plasticity in cultured hippocampal neurons. Moreover, when a sub-anesthetic dose of ketamine is given to mice, it increases synaptic GluA1 levels, but not GluA2, and GluA1 phosphorylation in the hippocampus within 1 hr after treatment. These changes are likely mediated by ketamine-induced reduction of calcineurin activity in the hippocampus. Using the open field and tail suspension tests, we demonstrate that a low dose of ketamine rapidly reduces anxiety-like and depression-like behaviors in both male and female mice. However, when in vivo treatment of a CP-AMPAR antagonist abolishes the ketamine's effects on animals' behaviors. We thus discover that ketamine at the low dose promotes the expression of CP-AMPARs via reduction of calcineurin activity, which in turn enhances synaptic strength to induce rapid antidepressant actions.

*For correspondence: seonil.kim@colostate.edu

†These authors contributed equally to this work

Competing interest: The authors declare that no competing interests exist.

## Editor's evaluation

This paper addresses an important clinical concern which is how the antidepressant ketamine exerts its effects acts rapidly. The authors suggest the reason is that ketamine increases glutamatergic transmission in the hippocampus. The strengths are the data are very good, and the limitations are discussed well.

## Introduction

Major depressive disorder (MDD), also referred to as clinical depression, is a severe mood disorder with a large global prevalence (*Diseases and Injuries, 2020*). When depression co-occurs with chronic

medical illnesses, untreated depression is linked to a lower quality of life, a higher risk of suicide, and impaired physical well-being (*Moussavi et al., 2007*; *Daly et al., 2010*; *Rihmer and Gonda, 2012*). As such, it is understandable why MDD represents a serious public health concern. Many antidepressant drugs have been used by targeting the monoamine systems to increase the amount of serotonin or norepinephrine in the brain (*Berton and Nestler, 2006*). However, it can take weeks or months for traditional antidepressants to fully manifest their therapeutic advantages (*Katz et al., 2004*). Moreover, less than 50% of all patients with depression have full remission with optimum treatment, thus there is still a great need for rapid medicinal relief to treat MDD (*Berton and Nestler, 2006*).

Over the past 50 years, the use of ketamine for anesthesia has become widespread in both human and veterinary medicine (*Kohtala, 2021*). Ketamine has also shown efficacy as a rapid-acting antidepressant only at low doses, particularly among those with treatment-resistant depression, while with increasing doses it evokes psychotomimetic actions and eventually produces anesthesia (*Abdallah et al., 2016*; *Miller et al., 2016*). Ketamine produces antidepressant effects within 1 hr after administration in humans (*Berman et al., 2000*; *Zarate et al., 2006*; *Liebrenz et al., 2009*). Notably, ketamine's half-life in the body is ~2 hours (*Autry et al., 2011*), but the ketamine's antidepressant effects last up to 1 week (*Berman et al., 2000*; *Zarate et al., 2006*; *Price et al., 2009*), strongly suggesting the involvement of neural plasticity (*Duman, 2018*). In fact, it is widely accepted that ketamine regulates a chain of molecular events connected with the facilitation of neural plasticity, including structural and functional plasticity, in the hippocampus and cortex, ultimately leading to the amelioration of depressive symptoms (*Kavalali and Monteggia, 2020*; *Kawatake-Kuno et al., 2021*; *Kohtala, 2021*; *Grieco et al., 2022*). Nonetheless, when, where, and how ketamine enhances the plasticity is still unclear (*Wu et al., 2021*). Therefore, our study aims to understand the mechanism underlying ketamine's rapid (less than an hour) antidepressant effects, which ultimately contributes to neural plasticity for long-term antidepressant benefits.

The main mechanism by which ketamine produces its therapeutic benefits on mood recovery is the promotion of neural plasticity in the hippocampus (*Miller et al., 2016*; *Ionescu et al., 2018*; *Aleksandrova et al., 2020*; *Kavalali and Monteggia, 2020*; *Grieco et al., 2022*). In fact, a recent study using the systematic and unbiased mapping approach that provides a comprehensive coverage of all brain regions discovers that ketamine selectively targets the hippocampus (*Davoudian et al., 2023*). However, ketamine is a noncompetitive NMDA receptor (NMDAR) antagonist that inhibits excitatory synaptic transmission (*Anis et al., 1983*). By inhibiting glutamatergic NMDARs, ketamine promotes synaptic inhibition rather than excitation (*Harrison and Simmonds, 1985*). Moreover, NMDARs are major $Ca^{2+}$ channels in excitatory synapses (*Zarei and Dani, 1994*). This suggests that ketamine deactivates NMDAR-dependent $Ca^{2+}$ signaling pathway. However, an important aspect of ketamine's therapeutic efficacy is mediated by enhancing neuronal $Ca^{2+}$ signaling (*Ali et al., 2020*; *Lisek et al., 2020*). Taken together, the main mechanisms believed to underlie ketamine's antidepressant effects converge on enhancing glutamatergic activity and neuronal $Ca^{2+}$-dependent signaling in the hippocampus (*Miller et al., 2016*; *Aleksandrova et al., 2020*; *Kavalali and Monteggia, 2020*; *Kawatake-Kuno et al., 2021*). Due to this, it becomes a puzzling question as to how ketamine rapidly enhances glutamatergic activity and $Ca^{2+}$ signaling while blocking NMDARs in the hippocampus.

One prominent hypothesis to explain these paradoxical effects of ketamine is that it directly inhibits NMDARs on excitatory neurons, which induces a cell-autonomous form of homeostatic synaptic plasticity to increase excitatory synaptic activity onto these neurons (*Miller et al., 2016*; *Kavalali and Monteggia, 2020*). This synaptic homeostasis is a negative-feedback response employed to compensate for functional disturbances in neurons and expressed via the regulation of glutamatergic AMPA receptor (AMPAR) trafficking and synaptic expression (*Lee, 2012*; *Diering and Huganir, 2018*). Postmortem studies have reported reductions in the mRNA expression levels of AMPAR subunit GluA1 and GluA3, but not GluA2, in the hippocampus of patients with depression (*Duric et al., 2013*), suggesting that subtype-specific AMPAR decrease in the hippocampus is implicated in depression. Moreover, accumulating evidence suggests that the antidepressant effects of ketamine can be mediated by alterations in AMPAR functions (*Moghaddam et al., 1997*; *Maeng et al., 2008*; *Nosyreva et al., 2013*; *Koike and Chaki, 2014*; *El Iskandrani et al., 2015*; *Zanos et al., 2016*; *Chowdhury et al., 2017*). Interestingly, following ketamine treatment in animals, many studies find elevated levels of GluA1, particularly in the hippocampus, whereas the results of other subunits' expression are less consistent (*Li et al., 2010*; *Nosyreva et al., 2013*; *Koike and Chaki, 2014*; *Yang et al., 2016*; *Zanos*

*et al., 2016*; *Georgiou et al., 2022*). This suggests that subtype specific activation of AMPARs is crucial for ketamine's antidepressant actions. However, it is unknown how ketamine selectively affects AMPAR subtype-specific functions in the hippocampus.

There are two distinct types of AMPARs formed through combination of their subunits: $Ca^{2+}$-impermeable GluA2-containing AMPARs and $\underline{Ca}^{2+}$-$\underline{P}$ermeable, GluA2-lacking, and GluA1-containing $\underline{AMPARs}$ (CP-AMPARs) (*Isaac et al., 2007*; *Liu and Zukin, 2007*). Activity-dependent AMPAR trafficking has long been known to be regulated by the phosphorylation of the GluA1 subunit (*Diering and Huganir, 2018*). Phosphorylation of serine 845 (S845) in GluA1 promotes GluA1-containing AMPAR surface expression, whereas dephosphorylation of S845 is involved in receptor internalization (*Diering and Huganir, 2018*; *Sathler et al., 2021*). We have previously shown that a decrease in neuronal $Ca^{2+}$ activity reduces the activity of $Ca^{2+}$-dependent phosphatase calcineurin, increasing GluA1 S845 phosphorylation to induce synaptic expression of CP-AMPARs, a part of homeostatic synaptic plasticity (*Kim et al., 2014*). It is thus possible that ketamine can reduce postsynaptic $Ca^{2+}$ and calcineurin activity via NMDAR antagonism, which increases GluA1 S845 phosphorylation to induce CP-AMPAR expression and enhances glutamatergic synaptic transmission. Indeed, a prior study demonstrated that ketamine induces CP-AMPAR expression in spiny projection neurons in the nucleus accumbens, although the study did not examine whether this change resulted in antidepressant behaviors (*Skiteva et al., 2021*). However, it is uncertain whether GluA2-containing or GluA2-lacking AMPARs are inserted or removed from hippocampal synapses following ketamine administration. Here, using cultured mouse hippocampal neurons, we reveal that ketamine at the low dose induces CP-AMPAR expression via reduction of neuronal $Ca^{2+}$ and calcineurin activity. Moreover, a low dose of ketamine in mice significantly reduces calcineurin activity and increases synaptic GluA1 levels, but not GluA2, in the hippocampus. Most importantly, ketamine at the low dose induces antidepression-like behaviors in mice within 1 hr after treatment, which is completely abolished by specifically blocking CP-AMPARs. Therefore, we discover a new molecular mechanism of ketamine's rapid antidepressant actions in which ketamine at the low doses promotes the expression of CP-AMPARs via reduction of calcineurin activity within one hour after treatment, which in turn enhances synaptic strength to induce antidepressant effects.

## Results

### Ketamine treatment selectively increases GluA1-containing AMPAR surface expression by decreasing calcineurin activity in cultured mouse hippocampal neurons

A large body of studies has found increased levels of GluA1 in the hippocampus after ketamine treatment in rodents; however, the results for other subunits' expression are less reliable (*Li et al., 2010*; *Nosyreva et al., 2013*; *Koike and Chaki, 2014*; *Yang et al., 2016*; *Zanos et al., 2016*; *Georgiou et al., 2022*). This led us to examined whether ketamine treatment changed surface expression of AMPAR subunits in cultured mouse hippocampal neurons. We treated 14 days in vitro (DIV) cultured mouse hippocampal neurons with 1 μM ketamine, the estimated concentration in the human brain after intravenous infusion of the therapeutic dose (*Hartvig et al., 1995*), for 1 hr and measured surface expression of AMPAR subunit GluA1 and GluA2 using biotinylation and immunoblots as shown previously (*Kim et al., 2014*; *Kim et al., 2015b*; *Kim et al., 2015a*; *Sztukowski et al., 2018*; *Sun et al., 2019*; *Roberts et al., 2021*). We found that ketamine treatment selectively increased surface expression of GluA1 when compared to the control (CTRL) (CTRL, 1.000 and ketamine, 1.598±0.543, p=0.0039), but not GluA2 (CTRL, 1.000 and ketamine, 1.121±0.464, p=0.6498; *Figure 1a*). As GluA1 phosphorylation at serine 831 (pGluA1-S831) and serine 845 (pGluA1-S845) are known to regulate GluA1-containing AMPAR surface trafficking (*Diering and Huganir, 2018*), we examined pGluA1-S831 and pGluA1-S845 levels one hour after 1 μM ketamine treatment in 14 DIV cultured hippocampal neurons as shown previously (*Sathler et al., 2021*). We found significantly higher pGluA1-S831 (CTRL, 1.000 and ketamine, 1.967±0.488, p=0.0149) and pGluA1-S845 levels (CTRL, 1.000 and ketamine, 2.399±1.024, p=0.0051) in ketamine-treated neurons than in the control (CTRL) (*Figure 1b*). This shows that ketamine treatment in cultured hippocampal neurons selectively increases GluA1 surface expression by increasing pGluA1-S831 and pGluA1-S845, which is consistent with the previous

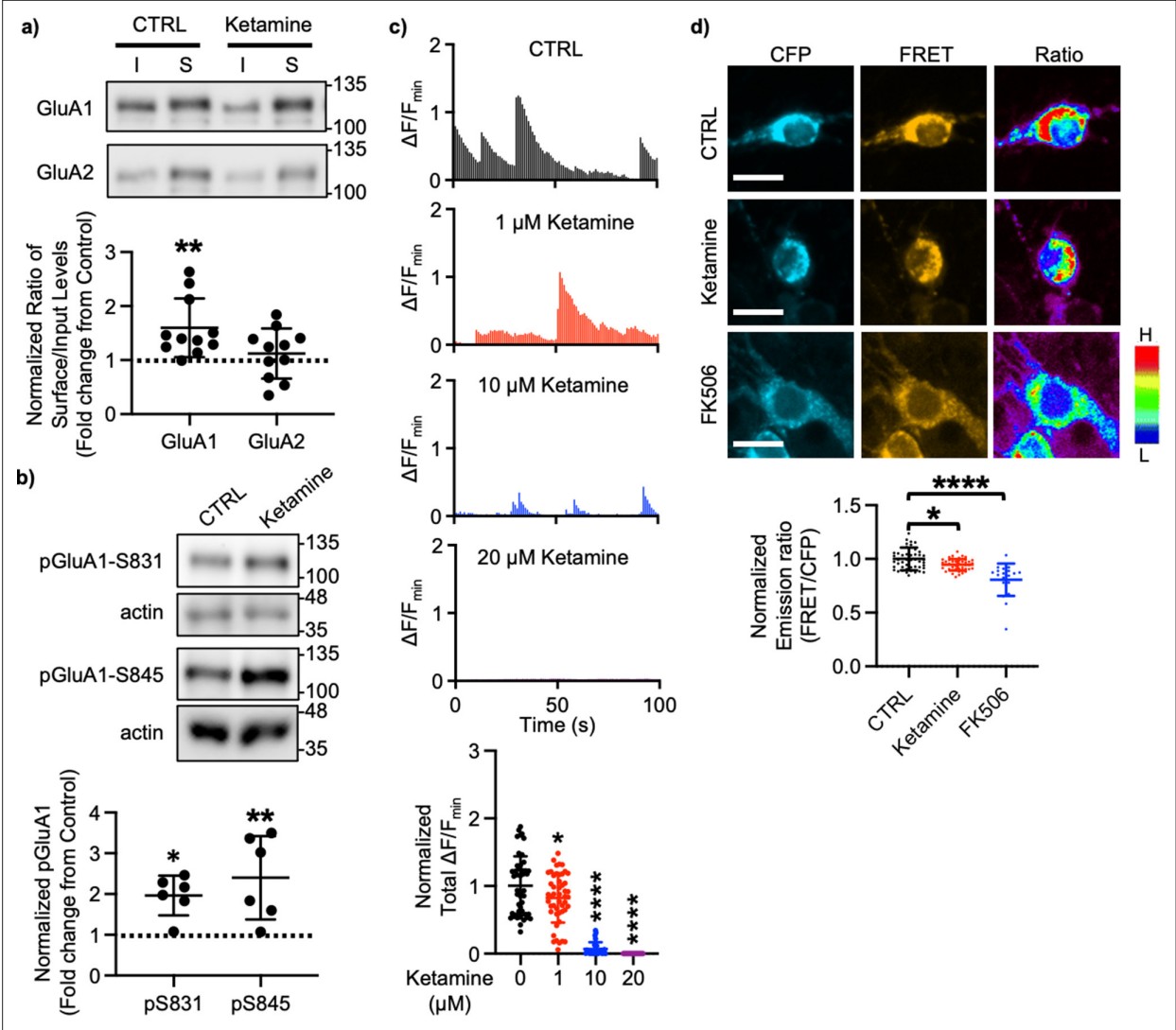

**Figure 1.** Ketamine treatment selectively increases GluA1-containing AMPAR surface expression by decreasing calcineurin activity in cultured mouse hippocampal neurons. (**a**) Representative immunoblots of input (I) and surface (S) levels in control (CTRL) and ketamine-treated neurons. Summary bar graphs of normalized surface GluA1 and GluA2 levels in each condition (n=11 immunoblots from 4 independent cultures, **p<0.01, the Kruskal-Wallis test with the Dunn's test). (**b**) Representative immunoblots of pGluA1 levels in control (CTRL) and ketamine-treated neurons. Summary graphs of normalized GluA1 phosphorylation levels in each condition (n=6 immunoblots from three independent cultures, *p<0.05 and **p<0.01, the Kruskal-Wallis test with the Dunn's test). (**c**) Representative traces of GCaMP7s signals in excitatory cells and summary data of normalized total $Ca^{2+}$ activity in each condition (n=number of neurons from two independent cultures, CTRL = 46, 1 µM Ketamine = 49, 10 µM Ketamine = 27, and 20 µM Ketamine = 26, *p<0.05 and ****p<0.0001, One-way ANOVA with the Tukey test). (**d**) Representative images of a CFP channel, a FRET channel, and a pseudocolored emission ratio (Y/C) in each condition [blue (L), low emission ratio; red (H), high emission ratio]. Scale bar is 10 µm. A summary graph showing average of emission ratio (Y/C) in each condition (n=number of cells, CTRL = 47, ketamine = 44, and FK506=20 from two independent cultures; *p<0.05 and ****p<0.0001; One-way ANOVA with the Tukey test). A scale bar indicates 10 µm. The position of molecular mass markers (kDa) is shown on the right of the blots. Mean ± SD.

The online version of this article includes the following source data for figure 1:

**Source data 1.** A compressed file containing images of (1) figures with the uncropped blots with the relevant bands labeled, (2) the original files of the full raw unedited blots, and (3) excel tables with the numerical data used to generate the *Figure 1a and b*.

findings showing crucial role of GluA1 phosphorylation in rapid antidepressant responses of ketamine (*Zhang et al., 2016*; *Zhang et al., 2017*; *Asim et al., 2022*).

We have previously shown that a decrease in $Ca^{2+}$-dependent phosphatase calcineurin activity significantly increases pGluA1-S845 and GluA1 surface expression (*Kim et al., 2014*). Previous studies have shown that 1 µM ketamine treatment can reduce ~50% of NMDA-induced currents (*Hare et al.,*

*2019*), whereas 10 µM is sufficient to block ~80% of NMDA-induced currents (*Halliwell et al., 1989*). Therefore, ketamine treatment can reduce neuronal $Ca^{2+}$ activity in the dosage-dependent manner, which in turn decreases calcineurin activity to elevate GluA1 phosphorylation and GluA1 surface expression. We thus examined whether ketamine treatment affected $Ca^{2+}$ activity in 14 DIV cultured hippocampal excitatory neurons using the previously described method with modification (*Kim et al., 2014*; *Kim et al., 2015b*; *Kim et al., 2015a*; *Sztukowski et al., 2018*; *Sun et al., 2019*; *Roberts et al., 2021*). For $Ca^{2+}$ imaging, a genetically encoded $Ca^{2+}$ indicator, GCaMP7s (*Dana et al., 2019*), was used to measure spontaneous somatic $Ca^{2+}$ activity in cultured hippocampal excitatory neurons in the presence of 1, 10, or 20 µM ketamine. We measured spontaneous $Ca^{2+}$ activity right after ketamine was treated, As consistent with the previous findings (*Halliwell et al., 1989*; *Hare et al., 2019*), we found a significant reduction in $Ca^{2+}$ activity in ketamine-treated neurons compared to control cells (CTRL) in the dosage-dependent manner (CTRL, 1.000±0.432 $F/F_{min}$, 1 µM ketamine, 0.820±0.363 $F/F_{min}$, p=0.036, 10 µM ketamine, 0.069±0.099 $F/F_{min}$, p<0.0001, and 20 µM ketamine, 0.000 $F/F_{min}$, p<0.0001) (*Figure 1c*). This demonstrates that ketamine treatment significantly reduces neuronal $Ca^{2+}$ activity in cultured hippocampal excitatory cells in the dosage-dependent manner.

To measure intracellular calcineurin activity directly, we used a Fluorescence Resonance Energy Transfer (FRET)-based calcineurin activity sensor as shown previously (*Kim et al., 2014*; *Mehta and Zhang, 2014*; *Kim et al., 2015b*; *Kim et al., 2015a*; *Sun et al., 2019*). We generated Sindbis virus to express the calcineurin activity sensor in cells (*Osten et al., 2000*). CFP, YFP, and FRET images in the soma of 14 DIV cultured hippocampal neurons were acquired 36 hr after infection, and the emission ratio was calculated as shown previously (*Kim et al., 2014*; *Kim et al., 2015b*; *Kim et al., 2015a*; *Sun et al., 2019*). We found that calcineurin activity was significantly decreased after one-hour 1 µM ketamine treatment compared to the control (CTRL) (CTRL, 1.000±0.106 and ketamine, 0.942±0.051, p=0.0170) (*Figure 1d*). Following one hour treatment of 5 µM FK506, a calcineurin inhibitor (*Liu et al., 1991*), calcineurin activity was markedly reduced compared to the control (CTRL; FK506, 0.806±0.150, p<0.0001) (*Figure 1d*) as shown previously (*Kim et al., 2014*). Taken together, ketamine-mediated NMDAR antagonism reduces neuronal $Ca^{2+}$ and calcineurin activity, which leads to a selective increase in GluA1 phosphorylation and GluA1-contraining AMPAR surface expression in cultured hippocampal neurons.

## Ketamine treatment induces CP-AMPAR expression to enhance glutamatergic activity and glutamate receptor plasticity in cultured mouse hippocampal neurons

We next examined how ketamine affected glutamatergic activity in cultured hippocampal excitatory neurons. Given that neuronal $Ca^{2+}$ is the secondary messenger responsible for transmitting depolarization status and synaptic activity (*Gleichmann and Mattson, 2011*), we carried out somatic $Ca^{2+}$ imaging with glutamate uncaging in cultured mouse hippocampal excitatory neurons to measure glutamatergic activity. We treated 14 DIV hippocampal cultures with 1 µM ketamine for 1 hr and measured glutamate-induced $Ca^{2+}$ signals. Glutamatergic activity was significantly higher in ketamine-treated neurons than control cells (CTRL) (CTRL, 1.000±0.278 $F/F_0$ and ketamine, 1.289±0.334 $F/F_0$, p=0.0022; *Figure 2a*). Because CP-AMPARs have larger single channel conductance (*Diering and Huganir, 2018*), we examined whether an increase in glutamatergic activity following ketamine treatment was mediated by CP-AMPARs. To do so, we treated hippocampal neurons with 1 µM ketamine for 1 hr and carried out $Ca^{2+}$ imaging with glutamate uncaging in the presence of 20 µM 1-naphthyl acetyl spermine (NASPM), a CP-AMPAR blocker. NASPM treatment was sufficient to abolish a ketamine-induced increase in glutamatergic activity (Ketamine +NASPM, 0.961±0.464 $F/F_0$, p=0.0006), while it had no effect on control cells (CTRL) (CTRL +NASPM, 0.965±0.350 $F/F_0$, p=0.9603) (*Figure 2a*). This shows that ketamine treatment induces CP-AMPAR expression and increases glutamatergic activity in cultured hippocampal excitatory neurons.

Ketamine's antidepressant effects are shown to be mediated by enhancing neural plasticity (*Grieco et al., 2022*). Specifically, ketamine enhances long-term potentiation (LTP) in the hippocampus, which contributes to antidepressant actions (*Yang et al., 2018a*; *Aleksandrova et al., 2020*). Moreover, CP-AMPARs can initiate LTP in the hippocampus particularly when NMDARs are blocked (*Jia et al., 1996*). We thus treated 14 DIV cultured hippocampal neurons with a glycine-based buffer, well-established to induce a form of chemically induced glutamate receptor-dependent LTP (cLTP) as shown

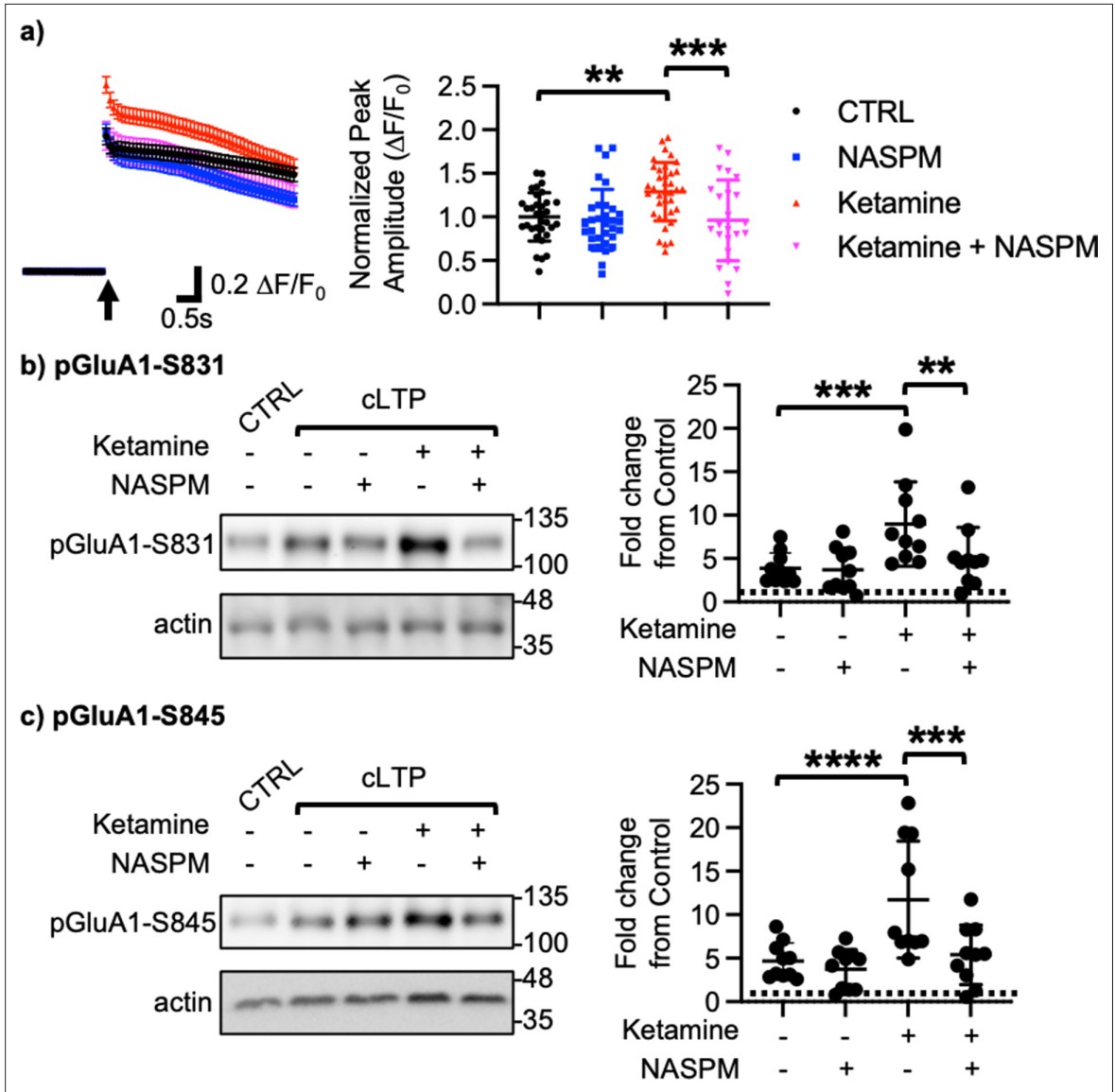

**Figure 2.** Ketamine treatment induces CP-AMPAR expression to enhance glutamatergic activity and glutamate receptor plasticity in cultured mouse hippocampal neurons. (**a**) Average traces of virally expressed GCaMP7s signals, and summary data of normalized peak amplitude in each condition (n=number of neurons, CTRL = 33, NASPM = 32, Ketamine = 37, and Ketamine +NASPM = 24 from two independent cultures; **p<0.01 and ***p<0.001; Two-way ANOVA with the Tukey test). An arrow indicates photostimulation. Representative immunoblots and quantitative analysis of (**b**) pGluA1-S831 and (**c**) pGluA1-S845 levels in each condition (n=10 immunoblots from five independent cultures, **p<0.01, ***p<0.001, and ****p<0.0001, the Kruskal-Wallis test with the Dunn's test). The position of molecular mass markers (kDa) is shown on the right of the blots. Mean ± SD.

The online version of this article includes the following source data for figure 2:

**Source data 1.** A compressed file containing images of (1) figures with the uncropped blots with the relevant bands labeled, (2) the original files of the full raw unedited blots, and (3) excel tables with the numerical data used to generate the *Figure 2b and c*.

previously (*Roberts et al., 2021*; *Sathler et al., 2021*) to examine whether ketamine enhanced gluta-mate receptor plasticity via the expression of CP-AMPARs. Following cLTP induction, pGluA1-S831 (CTRL, 1.000 and cLTP, 3.879±1.764, p=0.0027) and pGluA1-S845 levels (CTRL, 1.000 and cLTP, 4.658±2.090, p=0.0018) were significantly elevated in control neurons (CTRL), an indication of cLTP expression (*Figure 2b–c*). We next treated neurons with 1 µM ketamine for one hour, then induced cLTP, and measured GluA1 phosphorylation. When compared to neurons without ketamine treat-ment, pGluA1-S831 (Ketamine +cLTP, 8.978±4.861, p=0.0276) and pGluA1-S845 levels (Ketamine +cLTP, 11.73±6.717, p=0.0311) were significantly higher in ketamine-treated neurons, an indication

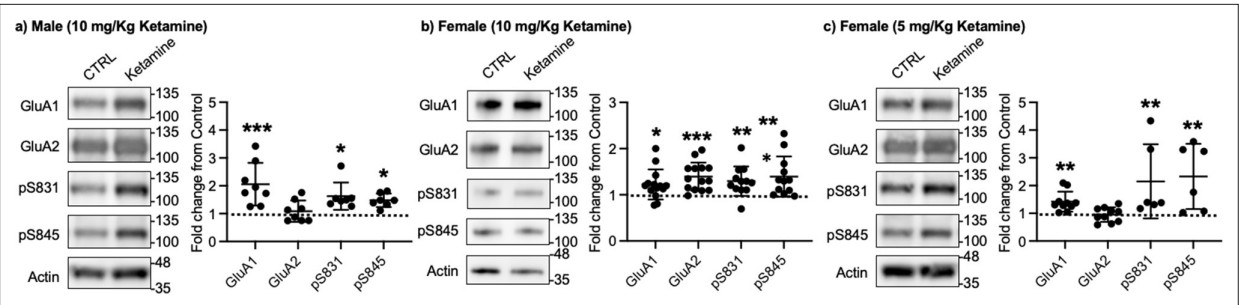

**Figure 3.** Synaptic GluA1 levels are selectively increased in the hippocampus following ketamine treatment. Representative immunoblots of AMPAR levels in the hippocampus of control (CTRL) and ketamine-treated (**a**) male (10 mg/Kg ketamine), (**b**) female (10 mg/kg ketamine), and (**c**) female (5 mg/Kg ketamine) mice. Summary graphs of normalized GluA1, GluA2, and GluA1 phosphorylation levels in each condition (n=number of immunoblots from 4 male and 3 female mice in each condition; Male (10 mg/Kg ketamine), CTRL = 8, GluA1=8, GluA2=8, pS831=7, and pS845=7, Females (10 mg/kg ketamine), CTRL = 14, GluA1=14, GluA2=14, pS831=12, and pS845=12, and Female (5 mg/Kg ketamine), CTRL = 10, GluA1=10, GluA2=10, pS831=6, and pS845=6, *p<0.05, **p<0.01, and ***p<0.001; the Kruskal-Wallis test with the Dunn's test). The position of molecular mass markers (kDa) is shown on the right of the blots. Mean ± SD.

The online version of this article includes the following source data for figure 3:

**Source data 1.** A compressed file containing images of (1) figures with the uncropped blots with the relevant bands labeled, (2) the original files of the full raw unedited blots, and (3) excel tables with the numerical data used to generate the *Figure 3a, b and c*.

of enhanced cLTP expression (*Figure 2b–c*). Importantly, a ketamine-induced increase in GluA1 phosphorylation was completely abolished when CP-AMPARs were blocked by treating neurons with 20 µM NASPM during cLTP (pGluA1-S831; Ketamine +cLTP + NASPM, 4.592±3.343, p=0.0299, and pGluA1-S845; Ketamine +cLTP + NASPM, 4.890±3.301, p=0.0279; *Figure 2b–c*). Notably, blocking CP-AMPARs had no effect on cLTP expression in the absence of ketamine treatment (pGluA1-S831; cLTP +NASPM, 3.684±2.503, p=0.7002, and pGluA1-S845; cLTP +NASPM, 3.724±2.275, p=0.4980; *Figure 2b–c*). Taken together, we demonstrate that ketamine enhances glutamate receptor plasticity via the expression of CP-AMPARs in cultured hippocampal cells.

## Synaptic GluA1 levels are selectively increased in the hippocampus following ketamine treatment

Given that ketamine selectively increases GluA1 phosphorylation and GluA1-containing AMPAR surface expression in cultured hippocampal neurons (*Figure 1a–b*), we examined whether ketamine treatment upregulates synaptic GluA1 and GluA2 levels in the mouse hippocampus. A low dose of ketamine (10 mg/kg), a condition that is shown to change hippocampal AMPAR expression in mice (*Zanos et al., 2016*), was intraperitoneally injected to 3-month-old male and female CD-1 mice, and saline was injected as a control. The postsynaptic density (PSD) fractions of the hippocampus were collected one hour after ketamine or saline injection, and synaptic GluA1, GluA2, pGluA1-S831, and pGluA1-S845 levels were measured by immunoblots as shown previously (*Kim et al., 2015b*; *Kim et al., 2015a*; *Farooq et al., 2017*; *Kim et al., 2018*). In male mice, we found GluA1 (CTRL, 1.000 and ketamine, 2.057±0.763, p=0.0005), pGluA1-S831 (CTRL, 1.000 and ketamine, 1.624±0.489, p=0.0158), and pGluA1-S845 levels (CTRL, 1.000 and ketamine, 1.480±0.243, p=0.0339), but not GluA2 levels (CTRL, 1.000 and ketamine, 1.088±0.383, p>0.9999), were significantly higher in the ketamine-treated hippocampal PSD fractions than the control (CTRL; *Figure 3a*). This shows that ketamine at the low dose significantly elevates synaptic GluA1 levels, which are likely mediated by increasing GluA1 phosphorylation in the male hippocampus, consistent with our findings in cultured hippocampal cells.

In contrast to male mice, 10 mg/kg ketamine injection in female mice significantly increased GluA1 (CTRL, 1.000 and ketamine, 1.224±0.324, p=0.0303), GluA2 (CTRL, 1.000 and ketamine, 1.393±0.304, p=0.0002), pGluA1-S831 (CTRL, 1.000 and ketamine, 1.296±0.319, p=0.0022), and pGluA1-S845 levels (CTRL, 1.000 and ketamine, 1.394±0.435, p=0.0014) in the PSD fractions (*Figure 3b*). This shows that ketamine at the low dose significantly increases both GluA1 and GluA2 levels in female hippocampal synapses. Interestingly, studies reveal that female rodents consistently respond to a lower dose of ketamine than male animals on depression-like behavioral tests, including forced swim

test and novelty suppressed feeding test (*Carrier and Kabbaj, 2013*; *Franceschelli et al., 2015*; *Zanos et al., 2016*; *Dossat et al., 2018*). We thus used a lower dose of ketamine (5 mg/kg) in female mice and examined synaptic GluA1 and GluA2 expression in the hippocampus to address if this sex difference in ketamine's effects on synaptic AMPAR expression in the hippocampus is dependent on ketamine concentration. The hippocampal PSD fractions were isolated one hour after 5 mg/kg ketamine or saline injection, and synaptic AMPAR levels were measured as shown above. Like male mice, synaptic GluA1 (CTRL, 1.000 and ketamine, 1.420±0.361, p=0.0053), pGluA1-S831 (CTRL, 1.000 and ketamine, 2.151±1.337, p=0.0014), and pGluA1-S845 levels (CTRL, 1.000 and ketamine, 2.330±1.177, p=0.0031) were significantly increased in the ketamine-treated female hippocampal synapses than the control, while GluA2 levels were not affected by ketamine (CTRL, 1.000 and ketamine, 0.952±0.260, p>0.9999) (*Figure 3c*). This demonstrates that a lower dose of ketamine (5 mg/kg) is sufficient to increase synaptic GluA1 levels by increasing GluA1 phosphorylation in the female hippocampus.

## Ketamine treatment significantly reduces anxiety-like behavior in mice, which requires CP-AMPARs

We next examined whether ketamine treatment affects anxiety-like behavior in mice using the open field test as shown previously (*Shou et al., 2019*). Ten mg/kg ketamine was intraperitoneally injected to 3-month-old male and female CD-1 mice, and saline was administered to a control. One hour after the injection, we measured total distance traveled (locomotor activity) and total time spent outside and inside (anxiety-like behavior) in the open field chamber. It has been shown that ketamine treatment in rodents induces hyperlocomotion and reduces anxiety-like behavior (*Hetzler and Wautlet, 1985*; *Irifune et al., 1991*; *Razoux et al., 2007*; *Chatterjee et al., 2011*; *de Araújo et al., 2011*; *Akillioglu et al., 2012*). Consistent with these findings, ketamine injection significantly increased total distance travelled compared to controls (CTRL) in male mice, an indication of hyperlocomotion (CTRL, 35.244±15.704 m and ketamine, 47.964±0.361 m, p=0.0382; *Figure 4a*). Furthermore, ketamine-treated male mice spent less time outside (CTRL, 1104.007±54.881 seconds and ketamine, 1048.647±50.779 seconds, p=0.0094) but more time inside (CTRL, 95.993±54.881 seconds and ketamine, 151.353±50.779, p=0.0094) than control mice, indicating decreased anxiety-like behavior (*Figure 4a*). To determine whether CP-AMPARs were required for these behavioral changes, we intraperitoneally administered 10 mg/kg IEM-1460, the concentration that is sufficient to show drug effects in animals (*Szczurowska and Mareš, 2015*; *Adotevi et al., 2020*), to ketamine-treated and saline-injected mice and performed the open field test one hour after drug injection. We found that blocking CP-AMPARs was sufficient to abolish the ketamine's effects in the open field test (Total distance travelled; Ketamine +IEM-1460, 33.069±10.000 m, p=0.0129, Time spent outside; Ketamine +IEM-1460, 1128.464±27.927 seconds, p=0.0001, and Time spent inside; Ketamine +IEM-1460, 71.536±27.927 seconds, p=0.0001) (*Figure 4a*). Conversely, IEM-1460 treatment had no effect on animals' behavior in the absence of ketamine (Total distance travelled; CTRL +IEM-1460, 30.782±13.867 m, p=0.8087, Time spent outside; CTRL +IEM-1460, 1134.809±44.172 seconds, p=0.3410, and Time spent inside; CTRL +IEM-1460, 65.191±44.172 seconds, p=0.3410) (*Figure 4a*). This shows that 10 mg/kg ketamine treatment significantly reduces male animals' anxiety-like behavior in the open field test, which is mediated by CP-AMPARs.

In female mice, unlike male animals, 10 mg/kg ketamine had no effect on locomotor activity (CTRL, 42.287±10.576 m and ketamine, 40.848±11.091 m, p=0.9804) (*Figure 4b*). However, like male mice, 10 mg/kg ketamine significantly reduced time spent outside (CTRL, 1103.561±24.289 seconds and ketamine, 1062.840±54.145 seconds, p=0.0330) but increased time spent inside (CTRL, 96.439±24.289 seconds and ketamine, 137.160±54.145, p=0.0330), an indication of reduced anxiety-like behavior (*Figure 4b*). To examine the role of CP-AMPARs in these behavioral changes, we intraperitoneally administered 10 mg/kg IEM-1460 and performed the open field test one hour after drug injection as described above. IEM-1460 injection had no effect on locomotor activity (Total distance travelled; CTRL +IEM-1460, 37.971±13.870 m, p=0.7456 and Ketamine +IEM-1460, 37.576±11.093 m, p=0.8239) (*Figure 4b*). However, in vivo CP-AMPAR inhibition was sufficient to reverse ketamine-induced behavioral changes (Time spent outside; Ketamine +IEM-1460, 1116.100±44.791 seconds, p=0.0033, and Time spent inside; Ketamine +IEM-1460, 83.900±44.791 seconds, p=0.0033) (*Figure 4b*). Like male mice, IEM-1460 treatment had no effect on behaviors in the open field test in the absence of

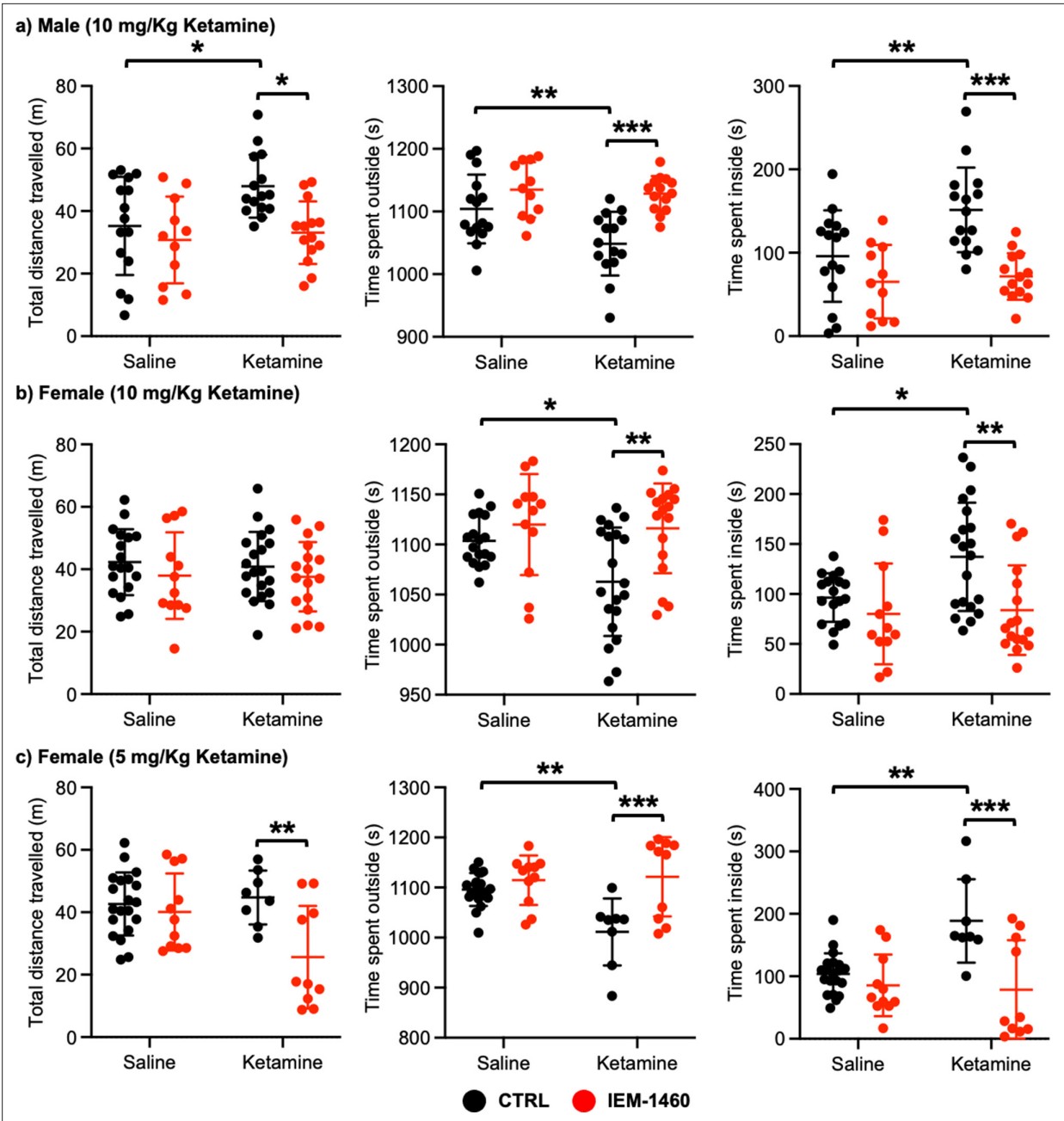

**Figure 4.** Ketamine treatment significantly reduces anxiety-like behavior in mice, which requires CP-AMPARs. The results of the open-field test measuring total distance travelled and time spent outside and inside in (**a**) male (10 mg/Kg ketamine), (**b**) females (10 mg/kg ketamine), and (**c**) females (5 mg/Kg ketamine) mice in each condition (n=number of mice, Male (10 mg/Kg ketamine); saline = 15, IEM 1460=11, Ketamine = 15, and Ketamine +IEM 1460=14, Female (10 mg/kg ketamine); saline = 18, IEM 1460=12, Ketamine = 20, and Ketamine +IEM 1460=17, and Female (5 mg/kg ketamine); saline = 20, IEM 1460=11, Ketamine = 8, and Ketamine +IEM 1460=10, *p<0.05, **p<0.01, and ***p<0.001, Two-way ANOVA with the Tukey test). Mean ± SD.

The online version of this article includes the following source data for figure 4:

**Source data 1.** A source data containing excel tables with the numerical data used to generate the *Figure 4a, b and c*.

ketamine (Total distance travelled; CTRL +IEM-1460, 37.971±13.870 m, p=0.7456, Time spent outside; CTRL +IEM-1460, 1119.917±50.469 seconds, p=0.7602, and Time spent inside; CTRL +IEM-1460, 80.083±50.469 seconds, p=0.7602) (*Figure 4b*). Given that a lower dose of ketamine (5 mg/kg) selectively increases synaptic GluA1 levels, but not GluA2, in the female hippocampus (*Figure 3c*), we intraperitoneally injected 5 mg/kg ketamine to 3-month-old female CD-1 mice and carried out the open

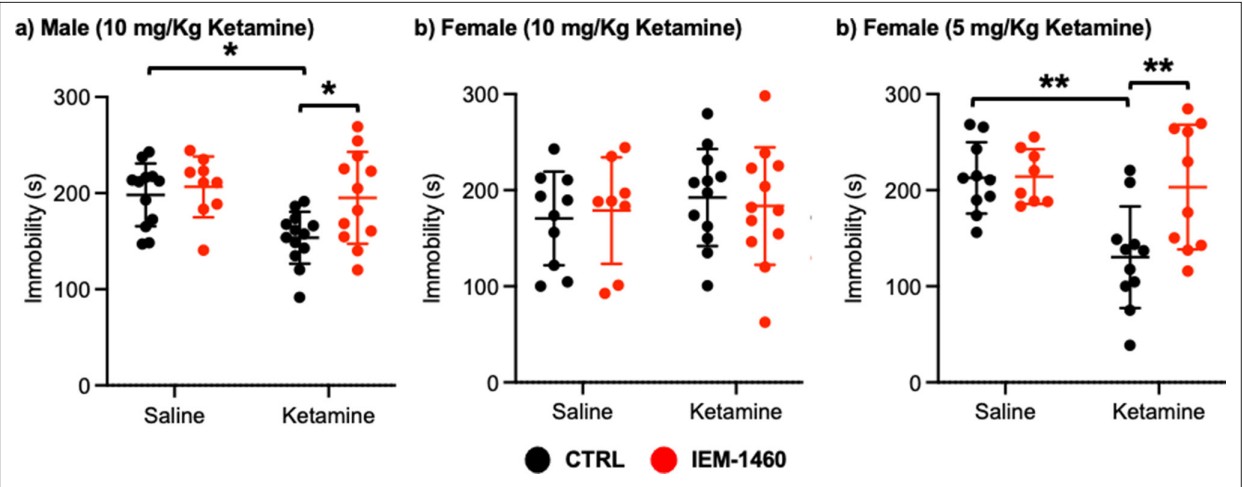

**Figure 5.** Ketamine treatment significantly reduces depression-like behavior in mice, which requires CP-AMPARs. The results of the tail suspension test measuring total immobility in (**a**) male (10 mg/Kg ketamine), (**b**) female (10 mg/kg ketamine), and (**c**) female (5 mg/Kg ketamine) mice in each condition (n=number of mice, Male (10 mg/Kg ketamine); saline = 12, IEM 1460=9, Ketamine = 13, and Ketamine +IEM 1460=12, Female (10 mg/kg ketamine); saline = 10, IEM 1460=8, Ketamine = 12, and Ketamine +IEM 1460=12, and Female (5 mg/Kg ketamine); saline = 10, IEM 1460=8, Ketamine = 11, and Ketamine +IEM 1460=10, *p<0.05 and **p<0.01, Two-way ANOVA with the Tukey test). Mean ± SD.

The online version of this article includes the following source data for figure 5:

**Source data 1.** A source data containing excel tables with the numerical data used to generate the *Figure 5a, b and c*.

field test as stated above. 5 mg/kg ketamine treatment in female mice was unable to increase loco-motor activity (CTRL, 42.648±10.103 m and ketamine, 44.736±8.647 m, p=0.9751), but CP-AMPAR inhibition significantly reduced locomotion only in ketamine-treated female mice (CTRL +IEM-1460, 40.098±12.326 m, p=0.9408, and ketamine +IEM-1460, 25.605±16.439 m, p=0.0080) (*Figure 4c*). This indicates that a lower dose of ketamine induces the expression of CP-AMPARs, contributing to locomotor activity in female mice. In addition, like 10 mg/kg ketamine treatment in male and female mice, 5 mg/kg ketamine injection in female animals significantly reduced anxiety-like behavior (Time spent outside; CTRL, 1096.190±32.931 seconds and Ketamine, 1011.288±66.949 seconds, p=0.0029, and Time spent inside; CTRL, 103.810±32.931 seconds and Ketamine, 188.713±66.949 seconds, p=0.0029), which was mediated by CP-AMPARs (Time spent outside; CTRL +IEM-1460, 1114.636±49.333 seconds, p=0.8026, and Ketamine +IEM-1460, 1121.520±79.288 seconds, p=0.0005, and Time spent inside; CTRL +IEM-1460, 85.364±49.333 seconds, p=0.8026, and Ketamine +IEM-1460, 78.480±79.288 seconds, p=0.0005) (*Figure 4c*). This indicates that 10 mg/kg and 5 mg/kg ketamine treatment in female mice significantly decrease anxiety-like behavior in the open-field test, which is dependent on CP-AMPARs.

## Ketamine treatment significantly reduces depression-like behavior in mice, which requires CP-AMPARs

We next used a tail suspension test as shown previously (*Kim et al., 2018*) to address whether ketamine-induced antidepressant actions were dependent on CP-AMPARs. Ten mg/kg ketamine was intraperitoneally injected to male and female 3-month-old CD-1 mice, and saline was administered to a control. As immobility in the tail suspension test is correlated with the depression-like state of the animals (*Kim et al., 2018*), we measured immobility and found that ketamine injection in male mice significantly decreased immobility relative to the control (CTRL) (CTRL, 198.183±32.632 seconds and ketamine, 153.569±54.145 seconds, p=0.0164), an indication of reduced depression-like behavior (*Figure 5a*). IEM-1460 treatment was sufficient to reverse ketamine-induced antidepressant effects on the tail suspension test (Ketamine +IEM-1460, 195.100±47.681 seconds, p=0.0285), while it had no effect on immobility in control animals (CTRL) (CTRL +IEM-1460, 206.578±31.598 seconds, p=0.9504) (*Figure 5a*). This demonstrates that 10 mg/kg ketamine treatment in male mice significantly reduces depression-like behavior, which is mediated by CP-AMPARs.

Unlike male animals, 10 mg/kg ketamine injection to female mice showed no antidepressant effects on our tail suspension test (CTRL, 170.550±48.708 seconds and ketamine, 192.433±50.495 seconds, p=0.7826) (*Figure 5b*). Moreover, IEM-1460 treatment was unable to affect depression-like behavior in female mice (CTRL +IEM-1460, 178.775±55.421 seconds, p=0.9885, and ketamine +IEM-1460, 183.517±61.042 seconds, p=0.9776) (*Figure 5b*). We thus treat 3-month-old female CD-1 mice with 5 mg/kg ketamine as described above and performed the tail suspension test. A lower dose of ketamine in female mice significantly decreased immobility in the tail suspension test (CTRL, 212.760±37.207 seconds and ketamine, 130.273±52.945 seconds, p=0.0026) (*Figure 5c*). Most importantly, CP-AMPAR antagonist treatment prevented ketamine's antidepressant effect (Ketamine +IEM-1460, 203.170±64.843 seconds, p=0.0087), while not influencing immobility in control animals (CTRL +IEM-1460, 214.050±28.469 seconds, p>0.9999) (*Figure 5c*). This shows that a lower dose of ketamine (5 mg/kg) is sufficient to induce antidepressant effects in female mice, which requires CP-AMPARs, like male animals.

## Ketamine treatment significantly reduces calcineurin activity in the hippocampus

Ketamine injection selectively increased synaptic GluA1 expression and its phosphorylation in the male and female hippocampus (*Figure 3*). We also revealed that CP-AMPARs were required for ketamine-induced antidepressant actions in male and female mice (*Figures 4 and 5*). Given our in vitro experiments showed that a ketamine-induced decrease in calcineurin activity played crucial roles in GluA1-containing AMPAR surface expression (*Figure 1c–d*), we virally expressed the FRET-based calcineurin activity sensor in the hippocampus to determine whether ketamine reduced calcineurin activity. To express the calcineurin activity sensor, Sindbis virus was stereotaxically injected into the hippocampal CA1 area of 3-month-old male and female CD-1 mice. Ketamine was intraperitoneally injected to animals 36 hr after the infection to ensure viral calcineurin activity sensor expression, and saline was administered as the control (CTRL). CFP, YFP, and FRET images were acquired in the soma of CA1 pyramidal neurons, and the emission ratio was calculated as shown in *Figure 1d*. We found that calcineurin activity in the male hippocampus was significantly decreased one-hour after 10 mg/kg ketamine treatment compared to the saline-treated hippocampus (CTRL) (CTRL, 1.000±0.382 and

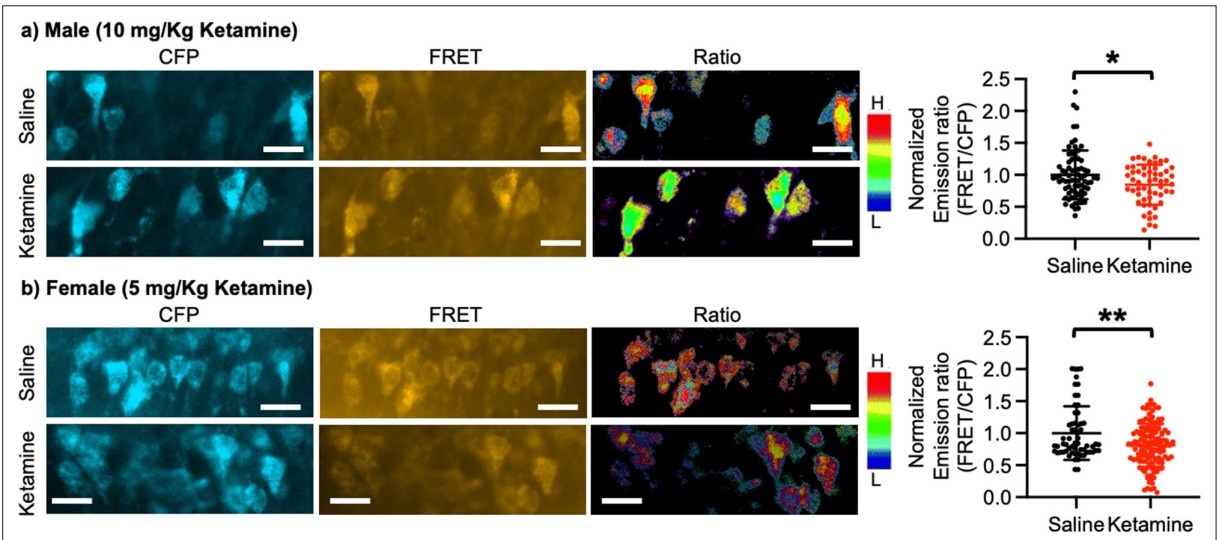

**Figure 6.** Ketamine treatment significantly reduces calcineurin activity in the hippocampus. Representative images of a CFP channel, a FRET channel, and a pseudocolored emission ratio (Y/C) in the (**a**) male and (**b**) female hippocampus in each condition [blue (L), low emission ratio; red (H), high emission ratio]. Scale bar is 10 µm. A summary graph showing average of emission ratio (FRET/CFP) in each condition n=number of cells [number of animals], Male; saline = 74 (*Ali et al., 2020*) and ketamine = 55 (*Akillioglu et al., 2012*), Female; saline = 61 (*Ali et al., 2020*) and ketamine = 130 (*Akillioglu and Karadepe, 2021*), *p<0.05 and **p<0.01, unpaired two-tailed student t-test. A scale bar indicates 25 µm. Mean ± SD.

The online version of this article includes the following source data for figure 6:

**Source data 1.** A source data containing excel tables with the numerical data used to generate the *Figure 6a and b*.

ketamine, 0.846±0.313, p=0.0164) (*Figure 6a*). We also injected 5 mg/kg ketamine to female mice and measured calcineurin activity in hippocampal CA1 neurons. Like the male hippocampus, 1-hr after 5 mg/kg ketamine treatment significantly reduced calcineurin activity in the female hippocampus (CTRL, 1.000±0.419 and ketamine, 0.812±0.338, p=0.0011) (*Figure 6b*). Therefore, ketamine treatment significantly reduces calcineurin activity in the hippocampus, which likely leads to an increase in GluA1 phosphorylation and the expression of CP-AMPARs. These glutamatergic changes in the hippocampus contribute to ketamine-induced antidepressant actions in animals.

## Discussion

Although an elevation of glutamatergic activity and neuronal $Ca^{2+}$-dependent signaling in the brain is thought to induce ketamine's antidepressant effects (*Miller et al., 2016*; *Aleksandrova et al., 2020*; *Kavalali and Monteggia, 2020*; *Kawatake-Kuno et al., 2021*), it is unclear how ketamine enhances these activities due to its nature of NMDAR antagonism. It has been suggested that ketamine's antidepressant effects are initiated by directly targeting NMDARs on excitatory neurons through a cell intrinsic mechanism (*Miller et al., 2016*). Ketamine can disrupt NMDAR basal activation on excitatory neurons. When synaptic excitation is reduced, a mechanism of homeostatic synaptic plasticity is activated, which causes an increase in excitatory synaptic responses in these neurons as a form of compensation (*Miller et al., 2016*). We and others have previously found that neuronal activity deprivation-induced homeostatic synaptic up-scaling can elevate glutamatergic synaptic activity and $Ca^{2+}$-dependent signaling via the expression of CP-AMPARs (*Thiagarajan et al., 2005*; *Kim et al., 2014*; *Sun et al., 2022*). CP-AMPARs could thus be an ideal candidate to counteract ketamine-induced NMDAR inhibition in neural plasticity and neuronal $Ca^{2+}$ signaling. In fact, studies in preclinical animal models have further demonstrated the necessity of AMPARs for the effects of ketamine, however their precise function is yet unknown (*Miller et al., 2016*). Moreover, multiple studies have shown that ketamine produces antidepressant effects within one hour after administration in humans (*Berman et al., 2000*; *Zarate et al., 2006*; *Liebrenz et al., 2009*) and rodents (*Maeng et al., 2008*; *Zanos et al., 2016*; *Fukumoto et al., 2017*). Therefore, the one-hour timeline is sufficient to show the antidepressant outcome. Additionally, a large volume of electrophysiological studies has demonstrated that ketamine affects synaptic activity within one hour (*Nosyreva et al., 2013*; *Zanos et al., 2016*; *Zhang et al., 2016*; *Widman and McMahon, 2018*; *Gerhard et al., 2020*). Here, our new findings demonstrate how ketamine rapidly (less than an hour) induces CP-AMPAR expression to adjust synaptic activity in the control of antidepressant behaviors.

Although a large group of AMPAR auxiliary subunits can provide heterogeneity of AMPAR trafficking (*Greger et al., 2017*), activity-dependent receptor trafficking has long been known to be regulated by the phosphorylation of GluA1 mainly in a two-step process (*Diering and Huganir, 2018*; *Pick and Ziff, 2018*). First, GluA1 S845 phosphorylation is mediated by cAMP-dependent protein kinase A (PKA) or cGMP-dependent protein kinase II (cGKII) (*Roche et al., 1996*; *Derkach et al., 2007*; *Serulle et al., 2007*). Importantly, GluA1 S845 phosphorylation promotes GluA1 surface expression and mediates LTP (*Banke et al., 2000*; *Ehlers, 2000*; *Lee et al., 2000*; *Esteban et al., 2003*; *Lee et al., 2003*; *Oh et al., 2006*; *Man et al., 2007*; *Diering et al., 2014*; *Kim et al., 2014*; *Kim et al., 2015b*; *Diering and Huganir, 2018*). In contrast, calcineurin-mediated dephosphorylation of GluA1 S845 is involved in receptor internalization (*Banke et al., 2000*; *Ehlers, 2000*; *Lee et al., 2000*; *Esteban et al., 2003*; *Lee et al., 2003*; *Oh et al., 2006*; *Man et al., 2007*; *Diering et al., 2014*; *Kim et al., 2014*; *Kim et al., 2015b*; *Diering and Huganir, 2018*). Second, when GluA1 is additionally phosphorylated at S831 by $Ca^{2+}$/calmodulin-dependent protein kinase II (CaMKII) or protein kinase C (PKC), and GluA1-containing AMPARs are targeted to the PSD, contributing to the enhanced synaptic transmission following LTP induction (*Barria et al., 1997*; *Derkach et al., 1999*; *Banke et al., 2000*; *Lee et al., 2000*; *Kristensen et al., 2011*; *Pick and Ziff, 2018*). Therefore, cooperative phosphorylation on GluA1 plays important roles in AMPAR trafficking and function in excitatory synapses (*Figure 7a*). Our new data suggest that ketamine-induced NMDAR antagonism significantly decreases neuronal $Ca^{2+}$ activity and subsequently calcineurin activity, leading to an increase in GluA1-containing, GluA2-lacking CP-AMPAR expression in the hippocampus via the elevation of GluA1 phosphorylation within one hour after ketamine treatment. Previous studies also demonstrate data consistent with our findings that GluA1 phosphorylation is crucial for ketamine-induced antidepressant effects in animals (*Zhang et al., 2016*; *Zhang et al., 2017*; *Asim et al., 2022*). These changes in glutamatergic synapses

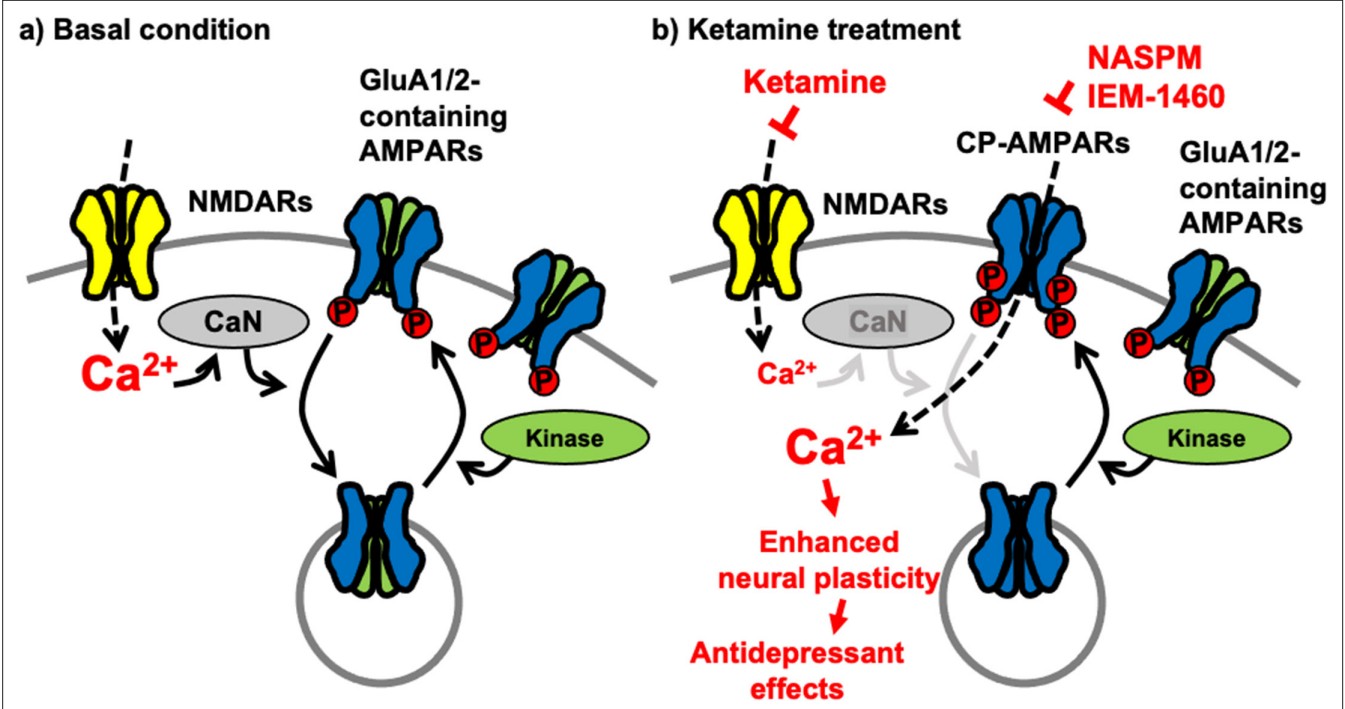

**Figure 7.** A schematic model of ketamine's antidepressant effects. (**a**) Under the basal conditions, stimulation of excitatory synapses results in $Ca^{2+}$ influx through glutamatergic NMDA receptors (NMDARs), which initiates intracellular pathways for neural plasticity. NMDAR-mediated $Ca^{2+}$ can activate calcineurin (CaN), a $Ca^{2+}$-dependent phosphatase that dephosphorylates the AMPA receptor (AMPAR) subunit GluA1, leading to AMPAR endocytosis. Several kinases, including PKA, cGKII, PKC, and CaMKII, on the other hand, can phosphorylate GluA1 to promote AMPAR surface expression. As a result, AMPAR trafficking and synaptic activity are controlled by the balance of kinases and phosphatases. (**b**) Because ketamine is a noncompetitive NMDAR antagonist, a therapeutic dose is enough to block NMDAR-mediated $Ca^{2+}$ influx in excitatory synapses. This can lower CaN activation and promote GluA1 phosphorylation, resulting in increased synaptic expression of GluA2-lacking, GluA1-containing $Ca^{2+}$-permeable AMPARs (CP-AMPARs). When ketamine is administered, CP-AMPAR-mediated $Ca^{2+}$ influx may replace NMDA-dependent $Ca^{2+}$ signaling. This increases neural plasticity, which leads to antidepressant benefits from ketamine.

enhances glutamate receptor plasticity in hippocampal neurons, which contributes to antidepressant effects in animals (*Figure 7b*). Taken together, we discover the molecular mechanisms of the ketamine-induced CP-AMPAR expression, which provides a better insight into the mechanisms that contributes to changes in neural plasticity and behaviors following ketamine treatment.

Although we and others show that GluA1 levels are selectively increased in the hippocampus within one hour after ketamine treatment in mice (*Li et al., 2010*; *Nosyreva et al., 2013*; *Koike and Chaki, 2014*; *Yang et al., 2016*; *Zanos et al., 2016*; *Georgiou et al., 2022*), other groups have also demonstrated an increase in both GluA1 and GluA2 levels following ketamine treatment (*Nosyreva et al., 2013*; *Zanos et al., 2016*). Interestingly, GluA1 and GluA2 levels were measured longer than 1 hr after ketamine treatment in these studies. It has been suggested that the insertion of CP-AMPARs in hippocampal synapses is only transient (less than one hour) following LTP induction (*Park et al., 2018*). Then, they are replaced by C̲a$^{2+}$-I̲mpermeable, GluA2-containing A̲MPARs (CI-AMPARs) because CP-AMPARs likely induce neurotoxicity via sustained synaptic $Ca^{2+}$ entry (*Noh et al., 2005*; *Dias et al., 2013*; *Park et al., 2018*). The discrepancy between our findings and those of others may therefore be due to the differences within ketamine treatment incubation time. In addition, other studies show no change in GluA1 and GluA2 levels after ketamine treatment (*Yao et al., 2018*; *Wojtas et al., 2022*). Notably, these studies examine ketamine effects in the frontal cortex or the meso-limbic circuit. Therefore, it is possible that ketamine may differentially affect glutamatergic synapses in different brain regions.

One significant discovery of our study is that, in contrast to male animals, female mice express CP-AMPARs after receiving a lower dose of ketamine, which promotes the antidepressant effects, an indication of enhanced ketamine antidepressant responses in female animals. Consistently, several studies using both male and female animals show an increased sensitivity to ketamine in females

(*Carrier and Kabbaj, 2013*; *Franceschelli et al., 2015*; *Zanos et al., 2016*; *Dossat et al., 2018*). One possible explanation of enhanced ketamine antidepressant responses in female rodents is different pharmacokinetics of ketamine in plasma and brain of the animals (*Saland and Kabbaj, 2018*). When compared to male rats, in female animals, higher concentrations of ketamine and norketamine, a ketamine's metabolite, are found in the medial prefrontal cortex and hippocampus over the 3 hr time course following treatment (*Saland and Kabbaj, 2018*). The study further demonstrates that longer half-lives and slower clearance rates in female rats contribute to greater effects of ketamine and its metabolites after treatment (*Saland and Kabbaj, 2018*). In addition, sex differences in the antidepressant activity of ketamine have shown to be mediated by sex hormones (*Carrier and Kabbaj, 2013*). Indeed, previous studies have been able to demonstrate a crucial role for ovarian hormones in the increased female behavioral sensitivity to low-dose ketamine (*Carrier and Kabbaj, 2013*; *Saland et al., 2016*). However, proestrus and diestrus female rats show no significant different pharmacokinetic profiles of ketamine, suggesting that sexual hormones have a stronger effect on ketamine downstream signaling pathways than the pharmacokinetic systems when it comes to causing sex-dependent behavioral sensitivity to ketamine (*Saland and Kabbaj, 2018*). Interestingly, ketamine and its two active metabolites, (*2 R,6R*)-hydroxynorketamine (HNK) and (*2 S,6S*)-HNK, can directly bind to estrogen receptor alpha (ERα) to increase GluA1 and GluA2 levels, an indication of AMPAR activation, which plays a key role in ketamine's antidepressant effects (*Ho et al., 2018*). Moreover, estradiol, the most potent and prevalent estrogen, is known to upregulate AMPAR functions by an increase in surface GluA2 levels (*Wei et al., 2014*; *Avila et al., 2017*). This is consistent with our findings in which both GluA1 and GluA2 expression is significantly increased in the female hippocampus when 10 mg/kg ketamine is injected (*Figure 3b*). Given that IEM-1460 treatment reverses anxiolytic behavior in female mice treated with 10 mg/kg ketamine (*Figure 4b*), this dose of ketamine induces the expression of both CP-AMPARs and CI-AMPARs in females. Additionally, these receptors likely contribute to the sex difference in ketamine-induced locomotor alteration between males and females, which is not surprising because multiple studies have already discussed the sex differences in the hyperlocomotion caused by ketamine in rodents (*Thelen et al., 2016*; *McDougall et al., 2019*; *Crawford et al., 2020*). Finally, more women than men are diagnosed with depression (*Holden, 2005*; *Kessler et al., 2005*; *Steiner et al., 2005*), which has been explained by the sex differences in the brain's structure and function as well as by the presence of sexually dimorphic hormones (*Kessler et al., 2003*; *Cosgrove et al., 2008*). However, the potential mechanisms underlying sex differences in response to ketamine have been particularly understudied at this time. Therefore, further discussion of the sex differences in the antidepressant activity of ketamine is needed.

A common etiological element in the production of major depression in humans is exposure to significant and frequently chronic psychological stress or trauma (*Hosang et al., 2014*; *Bonde et al., 2016*). However, the results from studies in a variety of mice strains generally show that ketamine has similar antidepressant effects in naive animals rather than having opposing effects in the presence or absence of chronic stress (*Weston et al., 2021*), consistent with our current findings. Nonetheless, there are mixed reports on ketamine's effects in naïve controls (*Ma et al., 2013*; *Franceschelli et al., 2015*; *Dong et al., 2017*; *Browne et al., 2018*; *Zhang et al., 2018*). Moreover, a recent clinical study reveals that a single infusion of ketamine shows therapeutic effects in patients with treatment-resistant depression, while it induces depressive symptoms in healthy individuals (*Nugent et al., 2019*). This indicates the importance of stressed states in determining the brain response to ketamine. Therefore, valid animal models of ketamine-induced antidepressant treatment will benefit by exhibiting stress-dependent behavioral responses.

In-depth investigations into the precise mechanisms underlying ketamine's effects have significantly advanced our understanding of depression and sparked the development of new ideas in molecular and cellular neuropharmacology. However, many basic and clinical questions regarding ketamine's antidepressant effects remain unanswered (*Kohtala, 2021*). The main mechanism by which ketamine produce its therapeutic benefits on mood recovery is the enhancement of neural plasticity in the hippocampus (*Miller et al., 2016*; *Aleksandrova et al., 2020*; *Kavalali and Monteggia, 2020*; *Grieco et al., 2022*). However, ketamine is a noncompetitive NMDAR antagonist that inhibits excitatory synaptic transmission (*Anis et al., 1983*). Research suggests multiple potential mechanisms to explain these paradoxical effects. In addition to the mechanism we have presented here, ketamine acts via direct inhibition of NMDARs localized on inhibitory interneurons, leading to disinhibition of

excitatory neurons and a resultant rapid increase in glutamatergic synaptic activity to activate $Ca^{2+}$ signaling pathway in the prefrontal cortex (*Ali et al., 2020*; *Deyama and Duman, 2020*; *Gerhard et al., 2020*). This stimulates the brain-derived neurotrophic factor (BDNF) signaling pathway, which subsequently increases the translation and synthesis of synaptic proteins to enhance AMPAR-mediated synaptic plasticity (*Deyama and Duman, 2020*). However, it is not completely understood how ketamine selectively inhibits NMDARs on inhibitory cells, given that the receptors are expressed in other cell types, including excitatory neurons. Another potential explanation is a NMDAR inhibition-independent mechanism that is mediated by the ketamine metabolites lacking NMDAR inhibition properties (*Carrier and Kabbaj, 2013*; *Franceschelli et al., 2015*; *Zanos et al., 2016*). In fact, the results of many human treatment trials indicate that other NMDAR antagonists lack the antidepressant properties of ketamine (*Newport et al., 2015*), supports this hypothesis. However, the United States Food and Drug Administration (FDA) recently approved one NMDAR antagonist for MDD. The current study offers a new neurobiological basis for ketamine's actions that depend on the NMDAR inhibition-dependent elevation of GluA1-containing AMPAR trafficking, which is likely independent from the previous described mechanisms including the BDNF-induced protein synthesis-dependent (*Deyama and Duman, 2020*) or the NMDAR inhibition-independent pathway (*Carrier and Kabbaj, 2013*; *Franceschelli et al., 2015*; *Zanos et al., 2016*). Nonetheless, there are still many important questions surrounding the molecular mechanisms of ketamine's actions. Therefore, future research will be needed to increase our comprehension of the pharmacological and neurobiological mechanisms of ketamine in the treatment of psychiatric diseases by addressing these questions.

# Materials and methods

**Key resources table**

| Reagent type (species) or resource | Designation | Source or reference | Identifiers | Additional information |
|---|---|---|---|---|
| Strain, strain background (*musculus males and females*) | CD1(ICR) | Charles River Laboratories | Stock No: 022 | |
| Transfected construct (The *Escherichia coli* bacteriophage P1) | pENN.AAV.CamKII 0.4.Cre.SV40 | Addgene | Addgene plasmid # 105558-AAV1; RRID:Addgene_105558 | |
| Transfected construct (R. norvegicus (rat), G. gallus (chicken); A. victoria (jellyfish)) | pGP-AAV-CAG-FLEX-jGCaMP7s-WPRE | Addgene | Addgene plasmid # 104495-AAV1; RRID:Addgene_104495 | |
| Transfected construct (*H. sapiens* (human), Synthetic) | pSinRep5-pcDNA3-CaNAR | This paper | *Mehta and Zhang, 2014* | Sindbis construct to infect and express calcineurin activity biosensor |
| Antibody | Anti-GluR1-NT (NT) antibody, clone RH95 (Mouse monoclonal) | Millipore | Cat. # MAB2263 | WB (1:2000) |
| Antibody | Anti-GluA2 antibody [EPR18115] (Rabbit monoclonal) | Abcam | Cat. # ab206293 | WB (1:2000) |
| Antibody | Anti-phospho-GluR1 (Ser831) antibody, clone N453 (Rabbit polyclonal) | Millipore | Cat. # 04–823 | WB (1:1000) |
| Antibody | Anti-GluR1 Antibody, phosphoSer 845 (Rabbit polyclonal) | Millipore | Cat. # AB5849 | WB (1:1000) |
| Antibody | Anti-Actin antibody [ACTN05 (C4)] (Mouse monoclonal) | Abcam | Cat. # ab3280 | WB (1:2000) |
| Commercial kit | Sindbis Expression System | Invitrogen | Cat. #: K750-01 | |
| Commercial kit | Enhanced Chemiluminescence (ECL) | Thermo Fisher Scientific | Cat. #: PI34580 | |
| Chemical compound, drug | PDS Kit, Papain Vial | Worthington Biochemical Corp. | Cat. #: LK003176 | ≥100 units per vial |
| Chemical compound, media | Neurobasal Medium without phenol red | Thermo Fisher Scientific | Cat. #: 12348017 | |

*Continued on next page*

*Continued*

| Reagent type (species) or resource | Designation | Source or reference | Identifiers | Additional information |
|---|---|---|---|---|
| Chemical compound, supplement | B27 | Thermo Fisher Scientific | Cat. #: 17504044 | |
| Chemical compound, drug | Glutamax | Thermo Fisher Scientific | Cat. #: 35050061 | |
| Chemical compound, antibiotics | Penicillin/Streptomycin | Thermo Fisher Scientific | Cat. #: 15070063 | |
| Chemical compound, drug | Ketamine hydrochloride | VetOne | Cat. #: 510189 | |
| Chemical compound, drug | Urethane | Sigma | Cat. #: U2500 | 1.2 g/kg |
| Chemical compound, drug | FK506 | Tocris Bioscience | Cat. #: 3631 | 5 μM |
| Chemical compound, drug | Tetrodotoxin (TTX) | Abcam | Cat. #: ab120055 | 2 μM |
| Chemical compound, drug | 4-methoxy-7-nitroindolinyl (MNI)-caged L-glutamate | Tocris Bioscience | Cat. #: 1490 | 1 mM |
| Chemical compound, drug | 1-Naphthyl acetyl spermine trihydrochloride (NASPM) | Tocris Bioscience | Cat. #: 2766 | 20 μM |
| Chemical compound, drug | IEM-1460 | Tocris Bioscience | Cat. #: 1636 | 10 mg/kg |
| Chemical compound, biotin | Sulfo-NHS-SS-biotin | Thermo Fisher Scientific | Cat. #: PI21331 | |
| Chemical compound, beads | Streptavidin-coated beads | Thermo Fisher Scientific | Cat. #: PI53150 | |
| Chemical compound, drug | Strychnine hydrochloride | Tocris Bioscience | Cat. #: 2785 | 1 μM |
| Chemical compound, drug | (-)-Bicuculline methochloride | Tocris Bioscience | Cat. #: 0131 | 20 μM |
| Software | ANY-maze tracking program | Stoelting Co. | https://www.any-maze.com | |
| Software | Prism 9 | GraphPad | https://www.graphpad.com/features | |
| Software | CellSens | Olympus | https://www.olympus-lifescience.com/en/software/cellsens/ | |

## Animals

CD-1 mice were obtained from Charles River (022) and bred in the animal facility at Colorado State University (CSU). Animals were housed under 12:12 hr light/dark cycle. Three-month-old male and female CD-1 mice were used in the current study. CSU's Institutional Animal Care and Use Committee (IACUC) reviewed and approved the animal care and protocol (3408).

## Primary hippocampal neuronal culture

Postnatal day 0 (P0) male and female CD-1 pups were used to produce mouse hippocampal neuron cultures as shown previously (*Sztukowski et al., 2018*; *Sathler et al., 2021*; *Sathler et al., 2022*). Hippocampi were isolated from P0 CD-1 mouse brain tissues and digested with 10 U/mL papain (Worthington Biochemical Corp., LK003176). Mouse hippocampal neurons were plated on following poly lysine-coated dishes for each experiment - glass bottom dishes (500,000 cells) for Ca$^{2+}$ imaging and FRET analysis, and 6 cm dishes (2,000,000 cells) for biochemical experiments. Neurons were grown in Neurobasal Medium without phenol red (Thermo Fisher Scientific, 12348017) with B27 supplement (Thermo Fisher Scientific, 17504044), 0.5 mM Glutamax (Thermo Fisher Scientific, 35050061), and 1% penicillin/streptomycin (Thermo Fisher Scientific, 15070063). The previous study evaluates maturation, aging, and death of mouse cortical cultured neurons for 60 DIV, which demonstrates that synaptogenesis is prominent during the first 15 days and then synaptic markers remain stable through 60 DIV (*Lesuisse and Martin, 2002*). In particular, the levels of glutamate receptors, including AMPARs and NMDARs, increase to a maximum by 10–15 DIV and then remain unchanged through 60 DIV. This

indicates that 14 DIV neurons that we used here are mature cells, and their maturity is likely comparable to that of older neurons. In fact, 14 DIV neurons have been used in many groups. Additionally, our cultures are shown to contain excitatory and inhibitory cells (*Sun et al., 2019*; *Roberts et al., 2021*) as well as glia (*Kaech and Banker, 2006*).

## Reagents

Ketamine hydrochloride (VetOne, 510189) was used in both in vitro and in vivo experiments. For neuronal cultures, we used 1, 10, or 20 µM ketamine. For mice, 5 mg/kg or 10 mg/kg ketamine was intraperitoneally injected to 3-month-old male and female CD-1 mice. We have an approval from IACUC to use ketamine and have the United States Drug Enforcement Administration license to use ketamine for research purpose (DEA# RK0573863). A total of 1.2 g/kg urethane (Sigma, U2500) was used for anesthetizing mice for stereotaxic surgery. Five µM FK506 (Tocris Bioscience, 3631), a condition that significantly reduces neuronal calcineurin activity to increase GluA1 phosphorylation, which induces the expression of CP-AMPARs to elevate AMPAR-mediated synaptic activity (*Kim et al., 2014*), was used to inhibit calcineurin activity in cultured hippocampal excitatory neurons. Two µM tetrodotoxin (TTX) (Abcam, ab120055) was used to block spontaneous $Ca^{2+}$ activity in cultured hippocampal excitatory neurons. One mM 4-methoxy-7-nitroindolinyl (MNI)-caged L-glutamate (Tocris Bioscience, 1490) was added to the culture media for glutamate uncaging. A total of 20 µM 1-Naphthyl acetyl spermine trihydrochloride (NASPM, Tocris Bioscience, 2766), a condition that significantly reduces CP-AMPAR-mediated synaptic and $Ca^{2+}$ activity (*Kim et al., 2014*; *Kim et al., 2015b*), was used to block CP-AMPARs in cultured hippocampal excitatory neurons. 10 mg/kg IEM-1460 (Tocris Bioscience, 1636) was intraperitoneally injected to 3-month-old male and female CD-1 mice to inhibit in vivo CP-AMPAR activity because it is blood-brain barrier (BBB)-permeable (*Wiltgen et al., 2010*; *Szczurowska and Mareš, 2015*; *Adotevi et al., 2020*).

## Surface biotinylation

Surface biotinylation was performed according to the previous studies (*Kim et al., 2014*; *Kim et al., 2015b*; *Kim et al., 2015a*; *Sztukowski et al., 2018*; *Sun et al., 2019*; *Roberts et al., 2021*). Cells were washed with ice-cold PBS containing 1 mM $CaCl_2$ and 0.5 mM $MgCl_2$ and incubated with 1 mg/ml Sulfo-NHS-SS-biotin (Thermo Fisher Scientific, PI21331) for 15 min on ice. Following biotin incubation, neurons were washed with 20 mM glycine to remove the excess of biotin, and cells were lysed in 300 µl RIPA buffer for one hour. 10% of total protein was separated as input samples, and protein lysates were incubated overnight with streptavidin-coated beads (Thermo Fisher Scientific, PI53150) at 4 °C under constant rocking. The beads containing surface biotinylated proteins were separated by centrifugation. Biotinylated proteins were eluted from streptavidin beads with SDS loading buffer. Surface protein fractions and their corresponding total protein samples were analyzed by immunoblots.

## Chemical LTP (cLTP)

cLTP protocol was followed as previously described (*Diering et al., 2014*; *Roberts et al., 2021*; *Sathler et al., 2021*). Fourteen DIV hippocampal cultured neurons were washed three times in $Mg^{2+}$ free buffer 150 mM NaCl, 2 mM $CaCl_2$, 5 mM KCl, 10 mM HEPES, 30 mM glucose, 1 µM strychnine hydrochloride (Tocris Bioscience, 2785), and 20 µM (-)-Bicuculline methochloride (Tocris Bioscience, 0131) and incubated in glycine buffer ($Mg^{2+}$-free buffer with 0.2 mM glycine) at 37 °C for 5 min. Then, $Mg^{2+}$ buffer ($Mg^{2+}$-free buffer with 2 mM $MgCl_2$) was added to block NMDARs and cells were incubated at 37 °C for 30 min before being processed for immunoblots. To inhibit CP-AMPARs, we added 20 µM NASPM in glycine and $Mg^{2+}$ buffer.

## Immunoblots

Immunoblots were performed as described previously (*Kim et al., 2005*; *Kim et al., 2014*; *Kim et al., 2015c*; *Kim et al., 2015b*; *Kim et al., 2015a*; *Farooq et al., 2017*; *Kim et al., 2016*; *Kim et al., 2018*; *Shou et al., 2019*; *Sztukowski et al., 2018*; *Sun et al., 2019*; *Roberts et al., 2021*; *Sathler et al., 2021*; *Tran et al., 2021*; *Sathler et al., 2022*). The protein concentration in total cell lysates was determined by a BCA protein assay kit (Thermo Fisher Scientific, PI23227). Equal amounts of protein samples were loaded on 10% glycine-SDS-PAGE gel. The gels were transferred to nitrocellulose membranes.

The membranes were blocked (5% powdered milk) for 1 hr at room temperature, followed by overnight incubation with the primary antibodies at 4 C. The primary antibodies consisted of anti-GluA1 (Millipore, 1:2000, MAB2263), anti-GluA2 (Abcam, 1:2000, ab206293), anti-phosphorylated GluA1-S831 (Millipore, 1:1000, 04823MI), anti-phosphorylated GluA1-S845 (Millipore, 1:1000, AB5849MI), and anti-actin (Abcam, 1:2000, ab3280) antibodies. Membranes were subsequently incubated by secondary antibodies for 1 hr at room temperature and developed with Enhanced Chemiluminescence (ECL) (Thermo Fisher Scientific, PI34580). Protein bands were quantified using ImageJ (https://imagej.nih.gov/ij/).

## GCaMP Ca²⁺ imaging

We measured spontaneous $Ca^{2+}$ activity in cultured hippocampal excitatory neurons because It has been shown that networks of neurons in culture can produce spontaneous synchronized activity (*Cohen et al., 2008*). In fact, network activity emerges at 3–7 DIV independent of either ongoing excitatory or inhibitory synaptic activity and matures over the following several weeks in cultures (*Cohen et al., 2008*). Therefore, the somatic $Ca^{2+}$ signals we observed are from the spontaneous network activity in cultured cells. To do this, we infected 4 DIV neurons with adeno-associated virus (AAV) expressing CamK2a-Cre (Addgene #105558-AAV1) - pENN.AAV.CamKII 0.4.Cre.SV40 was a gift from James M. Wilson (Addgene plasmid #105558; http://n2t.net/addgene:105558; RRID:Addgene_105558) - and Cre-dependent GCaMP7s (Addgene# 104495-AAV1) - pGP-AAV-CAG-FLEX-jGCaMP7s-WPRE was a gift from Douglas Kim & GENIE Project (Addgene plasmid #104495; http://n2t.net/addgene:104495; RRID:Addgene_104495) - (*Dana et al., 2019*) because when AAVs of the same serotype are co-infected, many neurons are transduced by both viruses (*Kim et al., 2013*). We then measured $Ca^{2+}$ activity in the soma of 14 DIV cultured hippocampal excitatory neurons with a modification of the previously described method (*Kim et al., 2014*; *Kim et al., 2015b*; *Kim et al., 2015a*; *Sztukowski et al., 2018*; *Sun et al., 2019*; *Roberts et al., 2021*). Glass-bottom dishes were mounted on a temperature-controlled stage on an Olympus IX73 microscope and maintained at 37 C and 5% $CO_2$ using a Tokai-Hit heating stage and digital temperature and humidity controller. For GCaMP7s, the images were captured right after 1, 10, or 20 µM ketamine was added to the media with a 10ms exposure time and a total of 100 images were obtained with a one-second interval. $F_{min}$ was determined as the minimum fluorescence value during the imaging. Total $Ca^{2+}$ activity was obtained by 100 values of $F/F_{min} = (F_t - F_{min}) / F_{min}$ in each image, and values of $F/F_{min} < 0.1$ were rejected due to potential photobleaching. The average total $Ca^{2+}$ activity in the control group was used to normalize total $Ca^{2+}$ activity in each cell. The control group's average total $Ca^{2+}$ activity was compared to the experimental groups' average as described previously (*Kim et al., 2014*; *Kim et al., 2015b*; *Kim et al., 2015a*; *Sztukowski et al., 2018*; *Sun et al., 2019*; *Roberts et al., 2021*).

## GCaMP Ca²⁺ imaging with glutamate uncaging

We carried out $Ca^{2+}$ imaging with glutamate uncaging as shown previously (*Wild et al., 2019*) in cultured hippocampal neurons one hour after 1 µM ketamine treatment. In addition, we added 20 µM NASPM right before $Ca^{2+}$ imaging to inhibit CP-AMPARs. For glutamate uncaging, 1 mM 4-methoxy-7-nitroindolinyl (MNI)-caged L-glutamate was added to the culture media, and epi-illumination photolysis (390 nm, 0.12 mW/mm², 1ms) was used. Two2 µM TTX was added to prevent action potential-dependent network activity. A baseline average of 20 frames (50ms exposure) ($F_0$) were captured prior to glutamate uncaging, and 50 more frames (50ms exposure) were obtained after single photostimulation. The fractional change in fluorescence intensity relative to baseline ($F/F_0$) was calculated. The average peak amplitude in the control group was used to normalize the peak amplitude in each cell. The control group's average peak amplitude was compared to the experimental groups' average.

## Sindbis virus infection in cultured neurons

Sindbis virus expressing the calcineurin activity sensor was produced as described previously (*Osten et al., 2000*). Calcineurin activity sensor cDNA (CaNAR) (a gift from Jin Zhang at Johns Hopkins University) was subcloned into pSinRep5 vector. BHK cells were electroporated with RNA of pSinRep5–CaNAR according to Sindbis Expression System manual (Invitrogen, K750-01). The pseudovirion-containing medium was collected 24 hr after electroporation, and the titer for the construct was

tested empirically in neuronal cultures. To express the calcineurin activity sensor in cultured neurons, 14 DIV neurons were infected with a titer resulting in infection of 20% of neurons (typically 1 µl of α-MEM virus stock diluted in 600 µl conditioned neurobasal-B27 medium per glass-bottom dish). It has been shown previously that no apparent adverse effects on morphology of the infected neurons was observed for up to 3 days post-infection (*Osten et al., 2000*). Cells were treated with 1 µM ketamine and/or 20 µM NASPM or 5 µM FK506 for one hour 24 hr after infection and fixed to analyze calcineurin activity.

## Sindbis virus infection in the mouse hippocampus

We virally expressed the calcineurin activity sensor using bilateral stereotaxic injection in the mouse hippocampus. Animals (3-month-old male and female CD-1 mice) were anaesthetized with 1.2 g/kg urethane. Anesthetic depth was confirmed with pedal response (foot retraction, response to non-damaging pressure of footpads using tweezers), ear twitch responses, and respiratory rates. Animal temperature was maintained with heating pads or warming gel packs. Once it was confirmed that the mice were properly anesthetized, the surgical field of the head of mice was aseptically prepared (shaved and prepped with betadine and alcohol). Animals were then placed in a stereotaxic frame (Stoelting). A small incision of the scalp was made with a sterile #10 surgical blade. With the aid of stereotaxic mounting equipment, a small hole was drilled in the bone using a high-speed drill and a dental bone drill bit, which has been sterilized. When the dura was exposed, a small pin hole was made, and a sterile syringe to inject Sindbis virus expressing the calcineurin activity sensor (1 µl) was lowered to the hippocampal CA1 area (Bregma coordinates: AP: − 1.95 mm, ML:±1.12 mm, DV: − 1.20 mm). During surgery, anesthetic depth was monitored every 5 min using pedal responses and respiration rates. After surgery, animals were allowed to recover from the anesthesia before being returned to their cages, and their health was closely monitored. Mice received analgesic doses of buprenorphine every 12 hr for 1 day after surgery. Buprenorphine was delivered by subcutaneous injection (0.1 mg/kg). Mice were monitored for any of the following signs of prolonged discomfort and pain: aggressiveness, hunched posture, failure to groom, awkward gait, vocalization, greater or less tissue coloration, eye discoloration, abnormal activity (usually less), hesitancy to move (especially in response to startle), water consumption, or food intake. Because neurons in the brains are preferentially infected with Sindbis virus at 36 hr after infection (*Furuta et al., 2001*), 10 mg/kg ketamine and/or 10 mg/kg IEM-1460 was intraperitoneally injected to animals 36 hr after the infection to ensure viral calcineurin activity sensor expression, and saline was administered to controls. Brain tissues were isolated one hour after treatment, fixed, and sectioned at 40 µm by using a vibratome. Hippocampal sections in each mouse were used imaged for hippocampal calcineurin activity.

## FRET analysis

Calcineurin activity was determined by the FRET emission ratio as described previously (*Kim et al., 2014*). CFP, YFP, and FRET images were acquired in the soma, and the following formula was used to calculate the emission ratio: (*FRET channel emission intensity – FRET channel emission intensity of background) / (CFP channel emission intensity – CFP channel emission intensity of background*) as described previously (*Kim et al., 2014*; *Kim et al., 2015a*; *Sun et al., 2019*). The higher emission ratio indicates the higher calcineurin activity.

## Behavioral tests

Both the open field test and tail suspension test have long been used to determine animals' anxiety- and depression-like behaviors, respectively, in rodents (*Seibenhener and Wooten, 2015*; *Ueno et al., 2022*). Specifically, the open field test has been widely used to measure the ketamine effects on anxiety-like behavior in rodents (*Guarraci et al., 2018*; *Pitsikas et al., 2019*; *Shin et al., 2019*; *Akillioglu and Karadepe, 2021*; *Yang et al., 2022*; *Acevedo et al., 2023*). We thus measured locomotor activity and anxiety-like behavior using the open field test as carried out previously (*Shou et al., 2019*). The test mouse was first placed in the center of the open field chamber (40 W x 40 L x 40 H cm) for 5 min. Animals were then allowed to explore the chamber for 20 min. A 20x20 cm center square was defined as the inside zone. The tail suspension test has also been used to examine the ketamine effects on depression-like behavior in animals (*Fukumoto et al., 2017*; *Yang et al., 2018b*; *Ouyang et al., 2021*; *Rawat et al., 2022*; *Viktorov et al., 2022*). Studies suggest that the forced

swim test and the tail suspension test are based on the same principle: measurement of the duration of immobility when rodents are exposed to an inescapable situation (*Castagné et al., 2011*). Importantly, it has been suggested that the tail suspension test is more sensitive to antidepressant agents than the forced swim test because the animals remain immobile longer in the tail suspension test than the forced swim test (*Cryan et al., 2005*). We thus used the tail suspension test to examine depression-like behavior as described previously (*Kim et al., 2018*). The test mouse was suspended by its tails from a rod suspended 20 cm above the tabletop surface with adhesive tape placed 1 cm from the tip of the tail. Animals were immobile when they exhibited no body movement and hung passively for >3 seconds. The time during which mice remained immobile was quantified over a period of 6 min. Mice that successfully climbed their tails to escape were excluded from the analysis. The behavior was recorded by a video camera. Data were analyzed using the ANY-maze tracking program to acquire total traveled distance (locomotor activity) and time spent outside and inside (anxiety-like behavior) for the open-field test and immobility (depression-like behavior) for the tail suspension test. All behavior tests were blindly scored by more than two investigators. Additionally, because ketamine produces antidepressant effects within 1 hr after administration in humans (*Berman et al., 2000*; *Zarate et al., 2006*; *Liebrenz et al., 2009*), our study aims to understand the mechanism underlying ketamine's rapid (less than an hour) antidepressant effects. Given that sucrose preference test and the novelty suppressed feeding test need multiple days, it would not be suitable to achieve our goals.

## Statistical analysis

The Franklin A. Graybill Statistical Laboratory at CSU has been consulted for statistical analysis in the current study, including sample size determination, randomization, experiment conception and design, data analysis, and interpretation. We used the GraphPad Prism 9 software to determine statistical significance (set at $p < 0.05$). Grouped results of single comparisons were tested for normality with the Shapiro-Wilk normality or Kolmogorov-Smirnov test and analyzed using an unpaired two-tailed Student's t-test when data are normally distributed. Differences between multiple groups were assessed by N-way analysis of variance (ANOVA) with the Tukey test or nonparametric Kruskal-Wallis test with the Dunn's test. The graphs were presented as mean ±Standard Deviation (SD).

## Materials availability statement

All renewable materials generated by this study will be made available to qualified individuals upon request.

## Acknowledgements

We thank members of the Kim laboratory for their generous support. We appreciate thoughtful suggestion from Drs. Mike Tamkun, Sanghun Lee, and Bret Smith. This work is supported by Student Experiential Learning Grants and College Research Council Shared Research Program from CSU, the NIH/NCATS Colorado CTSA Grant (UL1 TR002535), the Boettcher Foundation's Webb-Waring Biomedical Research Program, BrightFocus Foundation, and the NIH grant (1R03AG072102).

## Additional information

### Funding

| Funder | Grant reference number | Author |
|---|---|---|
| Colorado State University | | Anastasiya Zaytseva<br>Evelina Bouckova<br>McKennon J Wiles<br>Madison H Wustrau<br>Isabella G Schmidt<br>Seonil Kim |
| Boettcher Foundation | | Seonil Kim |
| NIH/NCATS Colorado CTSA Grant | UL1 TR002535 | Seonil Kim |

| Funder | Grant reference number | Author |
| --- | --- | --- |
| NIA | R03AG072102 | Seonil Kim |
| BrightFocus Foundation | | Seonil Kim |

The funders had no role in study design, data collection and interpretation, or the decision to submit the work for publication.

## Author contributions

Anastasiya Zaytseva, Evelina Bouckova, McKennon J Wiles, Funding acquisition, Investigation, Writing - original draft, Writing - review and editing; Madison H Wustrau, Investigation, Writing - original draft, Writing - review and editing; Isabella G Schmidt, Conceptualization, Investigation, Writing - review and editing; Hadassah Mendez-Vazquez, Investigation, Writing - review and editing; Latika Khatri, Resources; Seonil Kim, Conceptualization, Data curation, Formal analysis, Supervision, Funding acquisition, Validation, Investigation, Visualization, Writing - original draft, Writing - review and editing

## Author ORCIDs

Seonil Kim http://orcid.org/0000-0002-0451-2180

## Ethics

This study was performed in strict accordance with the recommendations in the Guide for the Care and Use of Laboratory Animals of the National Institutes of Health. All of the animals were handled according to approved institutional animal care and use committee (IACUC) protocols (#3408) of Colorado State University. The protocol was approved by the Committee on the Ethics of Animal Experiments of Colorado State University. All surgery was performed under urethane anesthesia, and every effort was made to minimize suffering.

## Decision letter and Author response

Decision letter https://doi.org/10.7554/eLife.86022.sa1
Author response https://doi.org/10.7554/eLife.86022.sa2

# Additional files

## Supplementary files

• MDAR checklist

## Data availability

Source data files have been provided for Figures (Fig. 1-6) that contain the numerical data used to generate the figures.

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
