## [Editor Report]

This paper addresses an important clinical concern which is how the antidepressant ketamine exerts its effects acts rapidly. The authors suggest the reason is that ketamine increases glutamatergic transmission in the hippocampus. The strengths are the data are very good, and the limitations are discussed well.

---

## [Decision Letter]

**Decision letter after peer review:**

Thank you for submitting your article "Ketamine's rapid antidepressant effects are mediated by ca^2+^-permeable AMPA receptors in the hippocampus" for consideration by *eLife*. Your article has been reviewed by 3 peer reviewers, one of whom is a member of our Board of Reviewing Editors, and the evaluation has been overseen by Lu Chen as the Senior Editor. The following individual involved in the review of your submission has agreed to reveal their identity: Hey-Kyoung Lee (Reviewer #3).

Essential revisions:

1. The brain area specificity of the behavioral data needs to be addressed. Without local delivery to the hippocampus, the core claims that are made in the paper are not well-supported. The authors can address this by (1) revising the text and title. One possibility is that, based on a prior study showing CP-AMPAR regulation in the nucleus accumbent, similar mechanisms occur across multiple brain areas. (2) The authors could conduct an experiment with local delivery to the hippocampus.

This is very important, as there are numerous studies from multiple labs, using a broad battery of behaviors, demonstrating that local ketamine delivery to mPFC is necessary and sufficient to account for behavioral changes. The observation that the hippocampus alone can underlie such effects would be very important.

2. The behaviors that were tested do not make a strong case for an antidepressant effect. The authors could address this by (1) being more cautious in their interpretations or (2) conducting more behavioral tests.

3. The time course of the effect of ketamine. The in vivo data showing when ketamine exerts effects is slower than the clinical effects of ketamine. The authors should address this difference.

4. The age of cultures is young and might not be relevant to the individual receiving ketamine. The authors provide PSD data from adult mice that support at least the AMPAR regulation part of the signaling mechanism. Phosphorylation changes are consistent with calcineurin stories from cultures. But it would be valuable to be more cautious.

*Reviewer #2 (Recommendations for the authors):*

1. Discussion and acknowledgment of limitations at the minimum; preferably an extension to more mature cultures that help generalize findings across development.

2. Experimental evaluation of effects timing.

3. Rigorous evaluation and consistent dosing regimen across sexes for the in vivo experiments.

4. Discussion point.

5. Experimental extension to additional behavioral models of depressive-like states (anhedonia, learned helplessness, or others). While each has limitations, which should be explicitly acknowledged, this would help generalize and strengthen the observations and significance of the study.

6. Intraperitoneal injections cannot be used to localize primary effects in vivo to the hippocampus, which would require local delivery. No claims related to direct effects can be made without those critical experiments.

7. Clarification of the dose would be sufficient. If 1uM was really the dose, additional positive controls are needed, as this is orders of magnitude below conventionally used concentrations.

---

## [Author Response]

Essential revisions:1. The brain area specificity of the behavioral data needs to be addressed. Without local delivery to the hippocampus, the core claims that are made in the paper are not well-supported. The authors can address this by (1) revising the text and title. One possibility is that, based on a prior study showing CP-AMPAR regulation in the nucleus accumbent, similar mechanisms occur across multiple brain areas. 2) The authors could conduct an experiment with local delivery to the hippocampus.This is very important, as there are numerous studies from multiple labs, using a broad battery of behaviors, demonstrating that local ketamine delivery to mPFC is necessary and sufficient to account for behavioral changes. The observation that the hippocampus alone can underlie such effects would be very important.

We agree with this point. We have thus removed “the hippocampus” in the title and have further made equivalent revisions in the other parts of the revised manuscript.

2. The behaviors that were tested do not make a strong case for an antidepressant effect. The authors could address this by (1) being more cautious in their interpretations or (2) conducting more behavioral tests.

We agree with the Reviewer’s concerns. However, both the open field test and tail suspension test have long been used to determine animals’ anxiety- and depression-like behaviors, respectively, in rodents (Seibenhener and Wooten, 2015; Ueno et al., 2022). Specifically, the open field test has been widely used to measure the ketamine effects on anxiety-like behavior in rodents (Guarraci et al., 2018; Pitsikas et al., 2019; Shin et al., 2019; Akillioglu and Karadepe, 2021; Yang et al., 2022; Acevedo et al., 2023). The tail suspension test has also been used to examine the ketamine effects on depression-like behavior in animals (Fukumoto et al., 2017; Yang et al., 2018; Ouyang et al., 2021; Rawat et al., 2022; Viktorov et al., 2022). Studies suggest that the forced swim test and the tail suspension test are based on the same principle: measurement of the duration of immobility when rodents are exposed to an inescapable situation (Castagne et al., 2011). Importantly, it has been suggested that the tail suspension test is more sensitive to antidepressant agents than the forced swim test because the animal will remain immobile longer in the tail suspension test than the forced swim test (Cryan et al., 2005). We thus chose to use the tail suspension test as opposed to the forced swim test. This information has now been included in the revised manuscript. Additionally, because ketamine produces antidepressant effects within one hour after administration in humans (Berman et al., 2000; Zarate et al., 2006; Liebrenz et al., 2009), our study aims to understand the mechanism underlying ketamine's rapid (less than an hour) antidepressant effects. Given that sucrose preference test and the novelty suppressed feeding test need multiple days, it would not be suitable to achieve our goals.

3. The time course of the effect of ketamine. The in vivo data showing when ketamine exerts effects is slower than the clinical effects of ketamine. The authors should address this difference.

We thank the reviewer's critique. Multiple studies have shown that ketamine produces antidepressant effects within one hour after administration in humans (Berman et al., 2000; Zarate et al., 2006; Liebrenz et al., 2009) and rodents (Maeng et al., 2008; Zanos et al., 2016; Fukumoto et al., 2017). Therefore, the one-hour timeline is sufficient to show the antidepressant outcome. Additionally, a large volume of electrophysiological studies has demonstrated that ketamine affects synaptic activity within one hour (Nosyreva et al., 2013; Zanos et al., 2016; Zhang et al., 2016; Widman and McMahon, 2018; Gerhard et al., 2020). Our study thus aims to understand the mechanism underlying ketamine's rapid (less than an hour) antidepressant effects, which contributes to neural plasticity for long-term antidepressant benefits. This new clarification has now been included in the revised manuscript.

4. The age of cultures is young and might not be relevant to the individual receiving ketamine. The authors provide PSD data from adult mice that support at least the AMPAR regulation part of the signaling mechanism. Phosphorylation changes are consistent with calcineurin stories from cultures. But it would be valuable to be more cautious.

We agree with the Reviewer’s concerns. The study by Lesuisse and Martin (J of Neurobiol. 2001) evaluates maturation, aging, and death of mouse cortical cultured neurons for 60 days in vitro (DIV), which demonstrates that synaptogenesis is prominent during the first 15 days and then synaptic markers remain stable through DIV 60. In particular, the levels of glutamate receptors, including AMPARs and NMDARs, increase to a maximum by DIV 10-15 and then remain unchanged through DIV 60. This indicates that DIV 14 neurons that we used in our study are mature cells, and their maturity is likely comparable to that of older neurons. In fact, DIV 14 neurons have been used in many groups. Additionally, our cultures are shown to contain excitatory and inhibitory cells (Sun et al., 2019; Roberts et al., 2021) as well as glia (Kaech and Banker, 2006).

Reviewer #2 (Recommendations for the authors):1. Discussion and acknowledgment of limitations at the minimum; preferably an extension to more mature cultures that help generalize findings across development.

We agree with the Reviewer’s concerns. The study by Lesuisse and Martin (J of Neurobiol. 2001) evaluates maturation, aging, and death of mouse cortical cultured neurons for 60 days in vitro (DIV), which demonstrates that synaptogenesis is prominent during the first 15 days and then synaptic markers remain stable through DIV 60. In particular, the levels of glutamate receptors, including AMPARs and NMDARs, increase to a maximum by DIV 10-15 and then remain unchanged through DIV 60. This indicates that DIV 14 neurons that we used in our study are mature cells, and their maturity is likely comparable to that of older neurons. In fact, DIV 14 neurons have been used in many groups.

2. Experimental evaluation of effects timing.

We agree with the Reviewer’s concerns. Ketamine is shown to produce antidepressant effects within one hour after administration in humans (Berman et al., 2000; Zarate et al., 2006; Liebrenz et al., 2009). Notably, ketamine’s half-life in the body is ~ 2 hours (Autry et al., 2011), yet a single therapeutic dose of ketamine produces a rapid antidepressant response in patients with major depressive disorder, with effects lasting up to one week (Berman et al., 2000; Zarate et al., 2006; Price et al., 2009), strongly suggesting it induces neural plasticity (Duman, 2018). As the Reviewer points out, the primary goal of our current study is to understand the mechanism underlying ketamine's rapid (less than an hour) antidepressant effects, which ultimately contributes to neural plasticity for long-term antidepressant benefits. This new clarification has now been included in the revised manuscript.

3. Rigorous evaluation and consistent dosing regimen across sexes for the in vivo experiments.

In the previous submission, we had reported biochemical data in addition to behavioral outcomes to provide the justification of the sex differences in ketamine’s effects. Specifically, we discovered that 5 mg/kg ketamine in females significantly increased synaptic GluA1 but not GluA2 in the hippocampus, an indication of CP-AMPAR expression, which is consistent with the findings in the behavioral data. Conversely, ketamine at the higher dose in females elevated both GluA1 and GluA2. With behavioral data, this suggests that 10 mg/kg ketamine in females induces both GluA2-containing and GluA2-lacking AMPARs, resulting in partial antidepressant effects. Our findings of an increased sensitivity to ketamine in females are consistent with previous studies showing that stress-naïve female rodents respond to lower doses of ketamine than male animals on depression-like behavioral tests, including forced swim test and novelty suppressed feeding test (Carrier and Kabbaj, 2013; Franceschelli et al., 2015; Zanos et al., 2016; Dossat et al., 2018). Therefore, we chose to use the lower dose (5 mg/kg). We have now included this justification in the Result section instead of the Discussion section.

4. Discussion point.

Multiple studies have shown that ketamine produces antidepressant effects within one hour after administration in humans (Berman et al., 2000; Zarate et al., 2006; Liebrenz et al., 2009) and rodents (Maeng et al., 2008; Zanos et al., 2016; Fukumoto et al., 2017). Therefore, the one-hour timeline is sufficient to show the behavioral outcome. Additionally, a large volume of electrophysiological studies has demonstrated that ketamine affects synaptic activity within one hour (Nosyreva et al., 2013; Zanos et al., 2016; Zhang et al., 2016; Widman and McMahon, 2018; Gerhard et al., 2020), which further supports this timeline.

5. Experimental extension to additional behavioral models of depressive-like states (anhedonia, learned helplessness, or others). While each has limitations, which should be explicitly acknowledged, this would help generalize and strengthen the observations and significance of the study.

We agree with the Reviewer’s concern. However, both the open field test and tail suspension test have long been used to determine animals’ anxiety-like and depression-like behaviors, respectively, in rodents (Seibenhener and Wooten, 2015; Ueno et al., 2022). Specifically, the open field test has been widely used to measure the ketamine effects on anxiety-like behavior in rodents (Guarraci et al., 2018; Pitsikas et al., 2019; Shin et al., 2019; Akillioglu and Karadepe, 2021; Yang et al., 2022; Acevedo et al., 2023). The tail suspension test has also been used to examine the ketamine effects on depression-like behavior in animals (Fukumoto et al., 2017; Yang et al., 2018; Ouyang et al., 2021; Rawat et al., 2022; Viktorov et al., 2022). Studies suggest that the forced swim test and the tail suspension test are based on the same principle: measurement of immobility duration while rodents are exposed to an inescapable situation (Castagne et al., 2011). Importantly, it has been suggested that the tail suspension test is more sensitive to antidepressant agents than the forced swim test because the animal will remain immobile longer in the tail suspension test than the forced swim test (Cryan et al., 2005). For this reason, we chose to use the tail suspension test instead of the forced swim test. This information has now been included in the revised manuscript. Additionally, because ketamine produces antidepressant effects within one hour after administration in humans (Berman et al., 2000; Zarate et al., 2006; Liebrenz et al., 2009), our study aims to understand the mechanism underlying ketamine's rapid (less than an hour) antidepressant effects. Given that sucrose preference test and the novelty suppressed feeding test need multiple days, it would not be suitable to achieve our goals.

6. Intraperitoneal injections cannot be used to localize primary effects in vivo to the hippocampus, which would require local delivery. No claims related to direct effects can be made without those critical experiments.

We agree with this point. We have thus removed “the hippocampus” in the title and have further made equivalent revisions in the other parts of the revised manuscript.

7. Clarification of the dose would be sufficient. If 1uM was really the dose, additional positive controls are needed, as this is orders of magnitude below conventionally used concentrations.

In the prior submission, we erroneously stated a concentration of 1 μM. In fact, we used MNI-caged glutamate at 1 mM. We have now updated it in the revised manuscript.